# Feature Likelihood Divergence: Evaluating the Generalization of Generative Models Using Samples

**Marco Jiralerspong**[*]
Université de Montréal and Mila

**Avishek (Joey) Bose**
McGill University and Mila

**Ian Gemp**
Google Deepmind

**Chongli Qin**
Google Deepmind

**Yoram Bachrach**
Google Deepmind

**Gauthier Gidel**[†]
Université de Montréal and Mila

## Abstract

The past few years have seen impressive progress in the development of deep generative models capable of producing high-dimensional, complex, and photo-realistic data. However, current methods for evaluating such models remain incomplete: standard likelihood-based metrics do not always apply and rarely correlate with perceptual fidelity, while sample-based metrics, such as FID, are insensitive to overfitting, i.e., inability to generalize beyond the training set. To address these limitations, we propose a new metric called the Feature Likelihood Divergence (FLD), a parametric sample-based metric that uses density estimation to provide a comprehensive trichotomic evaluation accounting for novelty (i.e., different from the training samples), fidelity, and diversity of generated samples. We empirically demonstrate the ability of FLD to identify overfitting problem cases, even when previously proposed metrics fail. We also extensively evaluate FLD on various image datasets and model classes, demonstrating its ability to match intuitions of previous metrics like FID while offering a more comprehensive evaluation of generative models. Code is available at https://github.com/marcojira/fld.

## 1  Introduction

Generative modeling is one of the fastest-growing areas of deep learning, with success stories spanning the artificial intelligence spectrum [Karras et al., 2020, Brown et al., 2020, Wu et al., 2021, Rombach et al., 2022]. Despite the growth of applications—and unlike supervised or reinforcement learning—there is a lack of a clear consensus on an evaluation protocol in high-dimensional data regimes in which these models excel. In particular, the standard metric of evaluating log-likelihood of held-out test data [Bishop and Nasrabadi, 2006, Goodfellow et al., 2016, Murphy, 2022] fails to provide a meaningful evaluation signal due to its large variability between repetitions [Nowozin et al., 2016] and lack of direct correlation with sample fidelity [Theis et al., 2015].

Departing from pure likelihood-based evaluation, sample-based metrics offer appealing benefits such as being able to evaluate any generative model family via their generated samples. Furthermore, sample-based metrics such as Inception score (IS) [Salimans et al., 2016], Fréchet Inception distance (FID) [Heusel et al., 2017], precision, and recall [Lucic et al., 2018, Sajjadi et al., 2018] have been shown to correlate with sample quality, i.e. the perceptual visual quality of a generated sample, and the perceptual sample diversity. Despite being the current defacto gold standard, sample-based

---

[*]e-mail correspondence to `marco.jiralerspong@mila.quebec`

[†]Canada Cifar AI Chair

[‡]Our metric (FLD) was named Feature Likelihood Score (FLS) in the previous versions of this paper.

37th Conference on Neural Information Processing Systems (NeurIPS 2023).

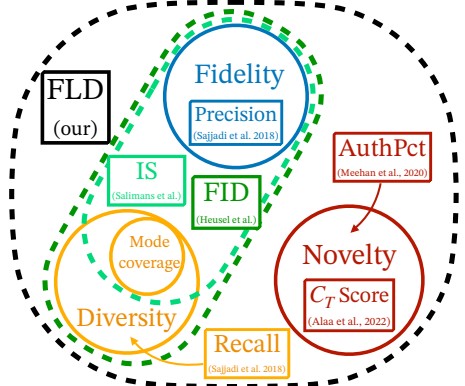

Figure 1: The generative model evaluation "trichotomy": fidelity, diversity and novelty. Each metric maps to a color delimiting its criteria for evaluation.

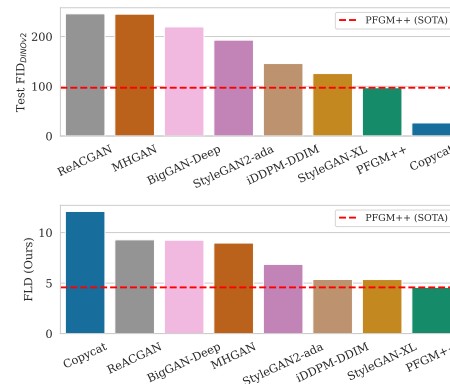

Figure 2: Test FID/FLD of various models on CIFAR10. FID values are higher than usual as we use the DINOv2 feature space with the test set as reference (10k samples) instead of the usual 50k.

metrics miss important facets of evaluation [Xu et al., 2018, Esteban et al., 2017, Meehan et al., 2020]. For example, on CIFAR10, the current standard FID computation uses 50k generated samples and 50k training samples from the dataset, a practice that does not take into account overfitting. Consider the worst-case scenario of a model called *copycat* which simply outputs copies of the training samples. Using the standard evaluation protocol, such a model would obtain a *FID of 0* (as we compare distances of two identical Gaussians)—a perfect score for a useless model. Instead, we could try looking at the FID of the copycat relative to the test set. While this improves the situation somewhat (see Fig. 2), copycat still obtains *better than SOTA test FID*, demonstrating that aiming for the lowest FID is highly vulnerable to overfitting.

While old generative models struggled to produce good quality samples, recent models have demonstrated the ability to memorize [Somepalli et al., 2022] and overfit [Yazici et al., 2020]. With the widespread adoption of deep generative models in high-stakes and industrial production environments, important concerns regarding data privacy [Carlini et al., 2023, Arora et al., 2018, Hitaj et al., 2017] should be raised. For instance, in safety-critical application domains such as precision medicine, data leakage in the form of memorization is unacceptable and severely limits the adoption of generative models—a few of which have been empirically shown to be guilty of "digital forgery" [Somepalli et al., 2022]. These concerns highlight the limitations with current evaluation metrics:

There are currently no *sample-based evaluation metrics* accounting for the trichotomy between sample *fidelity, diversity, and novelty* (Figure 1).

We believe this trichotomy encompasses what is required for a generative model to have the desired generalization properties, i.e., a "good" generative model should generate samples that are diverse and perceptually indistinguishable from the training data distribution, but *at the same time*, different from that training data. In other words, the more generated data looks like *unseen* test samples, the better. Additionally, by assessing the novelty of generated samples in relation to the training set, we can better identify potential privacy and copyright risks.

**Main Contribution**. We propose the feature likelihood divergence (FLD): a novel sample-based metric that captures sample fidelity, diversity, and novelty. FLD enjoys the same scalability as popular sample-based metrics such as FID and IS but crucially also assesses sample novelty, overfitting, and memorization. Evaluation using FLD has many consequential benefits:

1. **Explainability:** Samples that contribute the most (and the least) to the performance are identified.
2. **Diagnosing Overfitting:** As overfitting begins (i.e., copying of the training set) FLD identifies the copies and reports an inferior value *despite* no drop in sample fidelity and diversity.
3. **Holistic Evaluation:** FLD simultaneously is the only metric proposed in the literature that simultaneously evaluates the fidelity, diversity, and novelty of the samples (Fig. 1).
4. **Universal Applicability:** FLD applies to all generative models, including VAEs, Normalizing Flows, GANs, and Diffusion models with minimal overhead as it is computed only using samples.
5. **Flexibility:** Because of its connection with likelihood, FLD can be naturally extended to conditional and multi-modal generative modeling.

Intuitively, FLD achieves these goals by first mapping samples to a perceptually meaningful feature space such as a pre-trained Inception-v3 [Szegedy et al., 2016] or DINOv2 [Oquab et al., 2023]. Then, FLD is derived from the likelihood evaluation protocol that assesses the generalization performance of generative models in a similar manner to supervised learning setups. As most models lack explicit densities, we model the density of the generative model in our chosen feature space by using a mixture of isotropic Gaussians (MoG), whose means are the mapped features of the generated samples. We then fit the variances of the Gaussians to the train set in such a way that memorized samples obtain vanishingly small variances and thus worsen the density estimation of the MoG. Finally, we use the MoG and estimate the perceptual likelihood of some held-out test set.

## 2   Background and Related Work

Given a training dataset $\mathcal{D}_{\text{train}} = \{\mathbf{x}_i\}_{i=1}^n$ drawn from a distribution $p_{\text{data}}$, one of the key objectives of generative modeling is to train a parametric model $g$ that is able to generate novel synthetic yet high-quality samples—i.e., the distribution $p_g$ induced by the generator is close to $p_{\text{data}}$.[3]

**Likelihood Evaluation**. The most common metric, and perhaps most natural, is the negative log-likelihood (NLL) of the test set, whenever it is easily computable. While appealing theoretically, generative models typically do not provide a density (e.g. GANs) or it is only possible to compute a lower bound of the test NLL (e.g. VAEs, continuous diffusion models, etc.) [Burda et al., 2015, Song et al., 2021, Huang et al., 2021]. Even when possible, NLL-based evaluation suffers from a variety of pitfalls in high dimensions [Theis et al., 2015, Nowozin et al., 2016] and may often not correlate with higher sample quality [Nalisnick et al., 2018, Le Lan and Dinh, 2021]. Indeed many practitioners have empirically witnessed phenomena such as mode-dropping, mode-collapse, and overfitting [Yazici et al., 2020], all of which are not easily captured simply through the NLL.

**Sample-based Metrics**. As all deep generative models are capable of producing samples, an effective way to evaluate these models is via their samples. Such a strategy has the benefit of bypassing the need to compute the exact model density of a sample point—allowing for a unified evaluation setting. More precisely, given $\mathcal{D}_{\text{gen}} = \{\mathbf{x}^{\text{gen}}\}_{i=1}^m$ generated samples, where each $\mathbf{x}^{\text{gen}} \sim p_g$ and $\mathcal{D}_{\text{test}} = \{\mathbf{x}^{\text{test}}\}_{i=1}^n$ drawn from $p_{\text{data}}$, the goal is to evaluate how "good" the generated samples are with respect to the real data distribution. Historically, sample-based metrics for evaluating deep generative models have been based on two ideas: 1. using an Inception network [Szegedy et al., 2016] backbone $\varphi$ as a feature extractor to 2. compute a notion of distance (or similarity) between the generated and the real distribution. The Inception Score (IS) and the Fréchet Inception Distance (FID) are the two most popular examples and can be computed as follows:

$$\text{IS:} \quad e^{\frac{1}{m} \sum_{i=1}^m \text{KL}(p_\varphi(y|\mathbf{x}_i^{\text{gen}}) || p_d(y))}, \qquad \text{FID:} \quad \|\mu_g - \mu_p\|^2 + \text{Tr}(\Sigma_g + \Sigma_p - 2(\Sigma_g \Sigma_p)^{1/2})$$

where $p_\varphi(y|x)$ is the probability of each class given by the Inception network $\varphi$, $p_d(y)$ is the ratio of each class in the real data, $\mu_g := \frac{1}{m} \sum_{i=1}^m \varphi(\mathbf{x}_i^{\text{gen}}), \mu_p := \frac{1}{n} \sum_{i=1}^n \varphi(\mathbf{x}_i^{\text{test}})$ are the empirical means of each distribution, and $\Sigma_g := \frac{1}{m} \sum_{i=1}^m (\mathbf{x}_i^{\text{gen}} - \mu_g)(\mathbf{x}_i^{\text{gen}} - \mu_g)^\top, \Sigma_p := \frac{1}{n} \sum_{i=1}^n (\mathbf{x}_i^{\text{test}} - \mu_p)(\mathbf{x}_i^{\text{test}} - \mu_p)^\top$ are the empirical covariances.

The popularity of IS and FID as metrics for generative models is motivated by their correlation with perceptual quality, diversity, and ease of use. More recently, other metrics such as KID [Bińkowski et al., 2018] (an unbiased version of FID) and precision/recall (which disentangles sample quality and distribution coverage) [Sajjadi et al., 2018] have added nuance to generative model evaluation.

**Overfitting Evaluation**. Several approaches seek to provide metrics to detect overfitting and can be categorized based on whether one can extract an exact likelihood [van den Burg and Williams, 2021] or a lower bound to it via annealed importance sampling [Wu et al., 2016]. For GANs, popular approaches include training an additional discriminator in a Wasserstein GAN [Adlam et al., 2019] and adding a memorization score to the FID [Bai et al., 2021]. Alternate approaches include finding real data samples that are closest to generated samples via membership attacks [Liu et al., 2018, Webster et al., 2019]. Non-parametric tests have also been employed to detect memorization or exact data copying in generative models [Xu et al., 2018, Esteban et al., 2017, Meehan et al., 2020]. Parametric approaches to detect data copying have also been explored such as using neural network divergences [Gulrajani et al., 2020] or using latent recovery [Webster et al., 2019]. Finally, the $C_T$

---

[3]By close we mean either a divergence between distributions (e.g. KL, JSD) or a distance like Wasserstein.

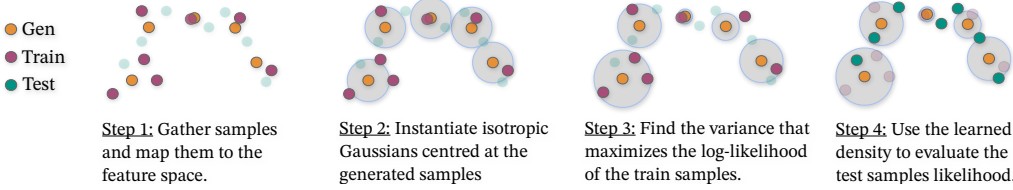

**Step 1:** Gather samples and map them to the feature space.

**Step 2:** Instantiate isotropic Gaussians centred at the generated samples

**Step 3:** Find the variance that maximizes the log-likelihood of the train samples.

**Step 4:** Use the learned density to evaluate the test samples likelihood.

Figure 3: Steps involved in our overfit mixture of Gaussians illustrated on a 2D example

[Meehan et al., 2020] test statistic and Alaa et al. [2022] proposes a multi-faceted metric with a binary sample-wise test to determine whether a sample is authentic (i.e., overfit).

## 3   Feature Likelihood Divergence

We now introduce our Feature Likelihood Score (FLD) which is predicated on the belief that a proper evaluation measure for generative models should go beyond sample quality and also inform practitioners of the generalization capabilities of their trained models. While previous sample-based methods have foregone density estimation in favor of computing distances between sample statistics, we seek to bring back a likelihood-based approach to evaluating generative models. To do so, we first propose our method for fitting a mixture of Gaussians (MoGs) to estimate the *perceptual* density of high-dimensional samples in a way that accounts for *overfitting*. Specifically, our method aims at attributing 1) a good NLL to high-quality, non-overfit images and 2) a poor NLL in cases of overfitting.

Intuitively, we say a generative model is overfitting if on average the distribution of generated samples is closer to the training set than the test set in feature space. We seek to characterize this exact behavior with FLD by a MoG where the variance parameter $\sigma^2$ of each Gaussian approaches zero around generated samples that are responsible for overfitting. In section 3.1 we deepen this intuition by outlining how the MoG used in FLD can be tuned to assess the perceptual likelihood of samples while punishing memorized samples, before giving a precise definition for memorization and overfitting under FLD in section 3.2.

### 3.1   Overfitting Mixtures of Gaussians

Our method consists of a simple sample-based density estimator amenable to a variety of data domains inspired by a traditional mixture of Gaussians (MoG) density estimator with a few key distinctions. Figure 3 summarizes the 4 key steps in computing FLD using our MoG density estimator (also detailed in Algorithm 1 in Appendix §D.1) which we now describe below. While it is indeed possible to train any other density model, a MoG offers a favorable tradeoff in being simple to use—while still being a universal density estimator [Nguyen et al., 2020]—and enjoying efficient scalability to large datasets.

**Step 1: Map to the feature space**. The first change we make is to use some map $\varphi$ to map inputs to some perceptually meaningful feature space. Natural choices for this include the representation space of Inception-v3 and DINOv2. While still high-dimensional, we ensure that a larger proportion of dimensions are useful and that the resulting $\ell_2$ distances between images are more meaningful.

**Step 2: Model the density using a MoG**. As in kernel density estimation (KDE), to estimate a density from some set of points $\mathcal{D}_{\text{gen}} = \{\mathbf{x}_j^{\text{gen}}\}_{j=1}^m$ we center an isotropic Gaussian around each point—i.e., the mean of the Gaussian is the coordinates of the point. This means that $j$-th data point has a Gaussian $\mathcal{N}(\varphi(\mathbf{x}_j^{\text{gen}}), \sigma_j^2 I_d)$. Then, to compute the likelihood of a new point $\mathbf{x}$, we simply calculate the mean likelihood assigned to that point by all Gaussians in the mixture:

$$p_\sigma(\mathbf{x}|\mathcal{D}_{\text{gen}}) := \frac{1}{m}\sum_{j=1}^m \mathcal{N}(\varphi(\mathbf{x})|\varphi(\mathbf{x}_j^{\text{gen}}), \sigma_j^2 I_d) = \frac{1}{m}\sum_{j=1}^m \frac{1}{(\sqrt{2\pi}\sigma_j)^d}\exp\left(\frac{-||\varphi(\mathbf{x}_j^{\text{gen}})-\varphi(\mathbf{x})||^2}{2\sigma_j^2}\right) \quad (1)$$

with the convention that $\mathcal{N}(\varphi(\mathbf{x})|\varphi(\mathbf{x}^{\text{gen}}), 0_d)$ is a dirac at $\varphi(\mathbf{x}^{\text{gen}})$. Henceforth, we denote this MoG estimator which has fixed centers initialized to a dataset (e.g. train set, generated set) as $\mathcal{N}(\varphi(\mathcal{D}); \Sigma)$, where $\Sigma$ is a diagonal matrix of bandwidths parameters—i.e. $\sigma^2 I$, where $\sigma^2$ is a vector.

**Step 3: Use the train set to select $\sigma_j^2$**. An important question in kernel density estimation is selecting an appropriate bandwidth $\sigma_j^2$. Overwhelmingly, a single bandwidth is selected which can either be derived statistically or by minimizing some loss through cross validation [Murphy, 2012]. We depart from this single bandwidth philosophy in favor of separate $\sigma_j^2$ values for each Gaussian. To select $\sigma_j^2$, instead of performing standard cross-validation on samples from $p_g$, we fit the bandwidths using

a subset of training examples $\{\varphi(\mathbf{x}_i^{\text{train}})\}_{i=1}^n$ by minimizing their negative log-likelihood (NLL). Specifically, we solve the following optimization problem:

$$\hat{\sigma}^2 \in \arg\max_{\sigma^2} \frac{1}{n} \sum_{i=1}^n \log\left(\frac{1}{m}\sum_{j=1}^m \frac{1}{(\sqrt{2\pi}\sigma_j)^d} \exp\left(\frac{-||\varphi(\mathbf{x}_j^{gen})-\varphi(\mathbf{x}_i^{train})||^2}{2\sigma_j^2}\right) + \mathcal{L}_i\right) \qquad (2)$$

where $\mathcal{L}_i$ is a base likelihood given to each sample (see Appendix §D for details). In particular, by centering each Gaussian on each $\mathbf{x}_j^{\text{gen}} \in \mathcal{D}_{\text{gen}}$ and fitting each $\sigma_j^2$ to the train set, we aim to have memorized samples obtain an overly small $\sigma_j^2$, worsening the quality of the MoG estimator (and thus penalizing memorization). The following proposition (proof in §F) formalizes this intuition.

**Proposition 1.** *Let $D_{ij} := ||\varphi(\mathbf{x}_j^{gen}) - \varphi(\mathbf{x}_i^{train})||^2$ be the distance between a generated sample and a train sample. Assume $\forall i,j : D_{ij} \leq \hat{D}$ with $\delta_j := \min_i D_{ij}$. Then, for any $l \in \{1,\dots,m\}$, we have that $\hat{\sigma}_l = O(\delta_l)$ where $\hat{\sigma}^2$ is a solution of Eq. 2.*

Proposition 1 implies that each element of the training set that has been memorized induces a Dirac in the MoG density Eq. 1. Thus, one can identify copies of training samples with the learned density. More generally, if one of the generated samples is unreasonably close to a training sample, its associated $\sigma^2$ will be very small as this maximizes the likelihood of the training sample. We illustrate this phenomenon with the Two-Moons dataset [Pedregosa et al., 2011] in Figure 4. Note that since this dataset is low-dimensional, we do not need to use a feature extractor (Step 1). In Figure 4 we can see that the more approximate copies of the training set appear in the generated set, the more the estimated density (using Eq. 2) contains high values around approximate copies of the training set. As such, overfitted generated samples yield an overfitted MoG that does not model the distribution of real data $p_{\text{data}}$ and will yield poor (i.e., low) log-likelihood on the test set $\mathcal{D}_{\text{test}}$.

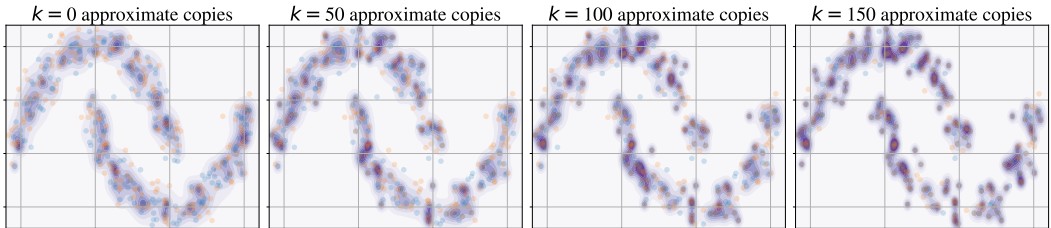

Figure 4: Estimated density (in purple) of the generated distribution using an MoG centered at the generated samples $\mathbf{x}_i^{\text{gen}}$ (in blue) Eq. 1. The selection of $\sigma_i^2$ is done via Eq. 2. The training points $\mathbf{x}_i^{\text{train}} \sim p_d$, sampled from the two-moons dataset, are represented in orange. The generated points correspond to $k$ approximates copies of the training set $\mathbf{x}_i^{\text{gen}} = \mathbf{x}_i^{\text{train}} + \mathcal{N}(0, 10^{-4})$, $i = 1,\dots,k$ and $200 - k$ independent samples from the data distribution $\mathbf{x}_i^{\text{gen}} \sim p_d, i = k+1,\dots,200$. The dark areas correspond to high-density values.

**Step 4: Evaluate MoG density**. A foundational concept used by FLD is to evaluate the perceptual negative log-likelihood of a held-out test set using an MoG density estimator. To quantitatively evaluate the density obtained in Step 3, we evaluate the negative log-likelihood of $\mathcal{D}_{\text{test}}$ under $p_{\hat{\sigma}}(\mathbf{x})$. As demonstrated in Figure 4, in settings with $k > 0$, the generated samples are too close to the training set, meaning that all test samples will have a high negative log-likelihood (as they are far from the center of Gaussians with low variances). Evaluation of the test set provides a succinct way of measuring the generalization performance of our generative model, which is a key aspect that is lost in metrics such as IS and FID. Our final FLD score is thus given by the following expression:

$$\text{FLD}(\mathcal{D}_{\text{test}}, \mathcal{D}_{\text{gen}}) := -\frac{100}{d}\log p_{\hat{\sigma}}(\mathcal{D}_{\text{test}}|\mathcal{D}_{\text{gen}}) - C, \qquad (3)$$

where $d$ is the dimension of the feature space $\varphi$ and is equivalent to looking at the $d^{th}$ root of the likelihood, and $C$ is a dataset dependant constant.[4] As a result of this adjustment by a constant, FLD is essentially estimating the forward Kullback-Leibler (KL) divergence (up to a constant factor) between the learned MoG distributions of the true data and the generated data. Higher FLD score values are indicative of problems in some of the three areas evaluated by FLD. Poor sample fidelity

---

[4]We set $C$ for the FLD to be zero with an ideal generator (i.e. a generator that could sample images from the true data distribution). In practice we can estimate such a $C$ by splitting the train set in 2, one part to represent the perfect $\mathcal{D}_{\text{gen}}$ and the other to compute $\hat{\sigma}$. Note that $C$ does not change the relative score between models.

leads to Gaussian centers that are far from the test set and thus a higher NLL. Similarly, a failure to sufficiently cover the data manifold will lead to some test samples yielding very high NLL. Finally, overfitting to the training set will yield a MoG density estimator that overfits and yield a poor NLL value on the test set.

**Computational Complexity of FLD**. To quantify the computational complexity of FLD we plot, in Figure 12 in §A.2, the computation time of various metrics with the number of samples. As depicted, we observe a linear scaling in the number of train samples for FLD which is in line with the computational cost of popular metrics like FID. Finally, the cost of computing FLD—and other metrics—is dwarfed by the cost of generating samples and then mapping them to an appropriate feature space which is a one-time cost and can be done prior to any metric computation.

## 3.2 Detecting Memorization and Overfitting

**Memorization**. One of the key advantages of FLD over other metrics like FID is that it can be used to precisely characterize memorization at the sample level. Intuitively, memorization occurs when a generated sample $\mathbf{x}_j^{\text{gen}}$ is an approximate copy of a training sample $\mathbf{x}_i^{\text{train}}$. By Proposition 1, such a phenomenon encourages the optimization of Eq. 2 to select a $\hat{\sigma}_j^2 \ll 1$ to achieve a high training NLL. As a result, such samples will assign a disproportionately high likelihood to that $\mathbf{x}_i^{\text{train}}$. To quantify this phenomenon, we compute the train likelihood assigned by each fitted Gaussian.

**Definition 3.1.** *Let $\delta > 0$. The sample $x_j^{gen}$ is said to be $\delta$-memorized if*

$$\mathcal{O}_j := \max_i \mathcal{N}(\varphi(x_i^{train})|\varphi(x_j^{gen}); \hat{\sigma}_j{}^2 I) > \delta \,. \tag{4}$$

The quantity $\mathcal{O}_j$ is appealing because it is efficient to compute, and the distribution of $\{\mathcal{O}_j\}$ allows us to quantify what is a "large" value for $\delta$ and identify the generated samples that are the most-likely copies of the training set. In §4.1.3, we explore this method to assess sample novelty.

**Overfitting**. Intuitively, a generative model is overfitting when it is more likely to generate samples closer to the train set than to the (unseen) test set. Thus, overfitting for deep generative models can be precisely defined using the standard tools for likelihood estimation.

**Definition 3.2.** *Given samples $\mathcal{D}_{gen} = \{\mathbf{x}^{gen}\}_{i=1}^n$ from a generative model $G$ trained on $\mathcal{D}_{train}$ and unseen test samples $\mathcal{D}_{test}$. We say that $G$ is overfitting if $\log p_{\hat{\sigma}}(\mathcal{D}_{test}|\mathcal{D}_{gen}) < \log p_{\hat{\sigma}}(\mathcal{D}_{train}|\mathcal{D}_{gen})$, i.e if:*

$$\text{Generalization Gap FLD} := \text{FLD}(\mathcal{D}_{train}, \mathcal{D}_{gen}) - \text{FLD}(\mathcal{D}_{test}, \mathcal{D}_{gen}) < 0 \tag{5}$$

For FLD, this effect is particularly noticeable due to the MoG being fit to the training set. In fact, samples that are too close to the train set relative to the test set have two effects: they worsen the density estimation of the MoG (increasing $\text{FLD}(\mathcal{D}_{\text{test}}, \mathcal{D}_{\text{gen}})$ and assign higher likelihood to the train set (lowering $\text{FLD}(\mathcal{D}_{\text{train}}, \mathcal{D}_{\text{gen}})$). In section 4.1.3 and Tab. 1 experiments we empirically validate the overfitting behavior of popular generative models using our above definition and also visualize samples that are most overfit.

**Evaluating individual sample fidelity**. While FLD focuses on estimating the density learned by the generative model, it is also possible to estimate the density of the data and use that to evaluate the likelihood of the generated samples[5]. In particular, instead of centering the Gaussians at the generated samples, we can instead **center them at the test set**. Then, after fitting this MoG to the train set, we compute the likelihood it assigns to generated samples and use that as a measure of sample quality. More formally:

$$\mathcal{Q}_j := \mathcal{N}(\varphi(\mathbf{x}_j^{\text{gen}})|\varphi(\mathcal{D}_{\text{test}}); \Sigma). \tag{6}$$

As is the case for $\mathcal{O}_j$, $\mathcal{Q}_j$ is easy to compute once the Mog is fit and can be used to rank and potentially filter out poor fidelity samples.

## 4 Experiments

We investigate the application of FLD on generative models that span a broad category of model families, including popular GAN and diffusion models. For datasets, we evaluate a variety of popular

---

[5]This is somewhat analogous to the difference between Recall and Precision [Sajjadi et al., 2018, Kynkään-niemi et al., 2019] where FLD can be seen as a smoother version of these metrics.

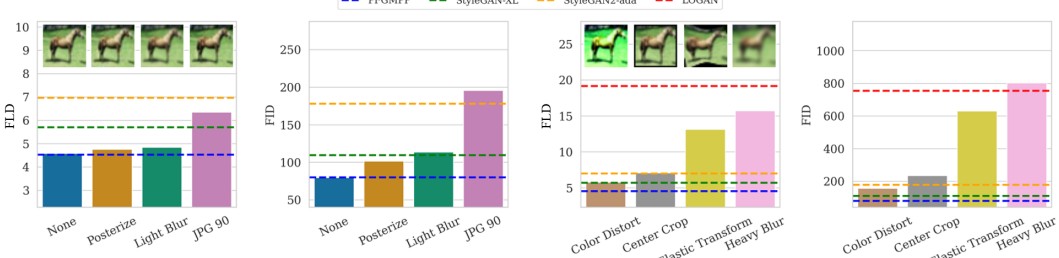

Figure 5: Starting from a set of SOTA samples produced by PFGM++, we replace each sample with a transformed copy. **Left:** Effect of nearly imperceptible transformations on FLD and FID (with corresponding values for various models as reference). **Right:** Effect of large transformations on FLD and FID.

natural image benchmarks in CIFAR10 [Krizhevsky et al., 2014], FFHQ [Karras et al., 2019] and ImageNet [Deng et al., 2009] . Through our experiments, we seek to validate the correlation between FLD and sample fidelity, diversity, and novelty.

Stein et al. [2023] find that the DINOv2 feature space allows for a more comprehensive evaluation of generative models (relative to Inception-V3) and correlates better with human judgement. As such, for our experiments, unless indicated otherwise, we map samples to the DINOv2 feature space [Oquab et al., 2023]. We do so even for other metrics (e.g. FID, Precision, Recall, etc.) which have typically used other feature spaces (comparisons with vanilla FID are provided in Appendix §C).

## 4.1 Trichotomic evaluation of FLD

We now experimentally validate FLD's ability to evaluate the samples of generative models along the three axes of fidelity, diversity and novelty.

### 4.1.1 Sample Fidelity

We evaluate the effect of 2 types of transformations on FLD and plot the results in Fig. 5. The first consists of minor, almost imperceptible transformations (very minor gaussian blur, posterizing and converting to a high quality JPG) that are problematic for FID. As described in [Parmar et al., 2022] small changes to images such as compression or the wrong form of anti-aliasing increase FID substantially despite yielding essentially indistinguishable samples. These transformations affect FLD but *noticeably less than FID*. This phenomenon occurs even when we use DINOv2 for both FLD and FID (though the effect is more drastic using the original Inception-V3 feature space, see Appendix §C).

Specifically, when all the samples are transformed, FID rates the imperceptibly transformed samples PFGM++ samples as worse than those produced by StyleGAN-XL (or even worse than StyleGAN2-ada). On the other hand, while the imperceptible transforms yield slightly worse FLD values (in part due to the feature space being sensitive to them), the FLD values are barely changed for the "Posterize" and "Light Blur" transforms and only somewhat worse for "JPG 90". The second type of transformations are larger transformations that affect the structure of the image (e.g. cropping, rotation, etc.) and have a significant negative impact on both FLD and FID.

### 4.1.2 Sample Diversity

We consider two experiments on CIFAR10 to evaluate the effect of mode coverage and diversity on FLD. For the first, we vary the number of classes included in the set of generated samples for various conditional generative models. For the second, instead of the original $n$ generated samples, we look at a set comprised of $\frac{n}{k}$ samples from the original set combined with $(k-1)$ approximate copies of each of those samples (i.e. for a total of $n$ samples). In both cases, we find a strong and consistent relationship (across all models) indicating that both sample diversity and mode coverage are important to get good FLD values and report our findings in Fig. 6.

### 4.1.3 Sample Novelty

We now study the effect of data copying on CIFAR10 on evaluation metrics for generative models. As overfitting can be more subtle than direct copying of the training set, we also consider transformed

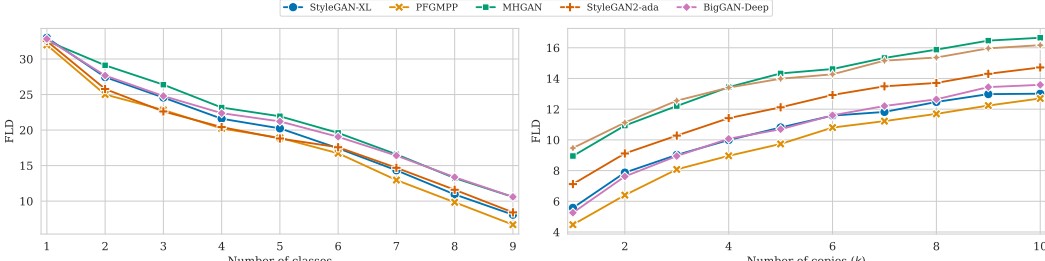

Figure 6: **Left:** FLD for various models as we increase the number of classes included in the generated samples. **Right:** FLD as the set of generated samples includes increasing amounts of copies of itself.

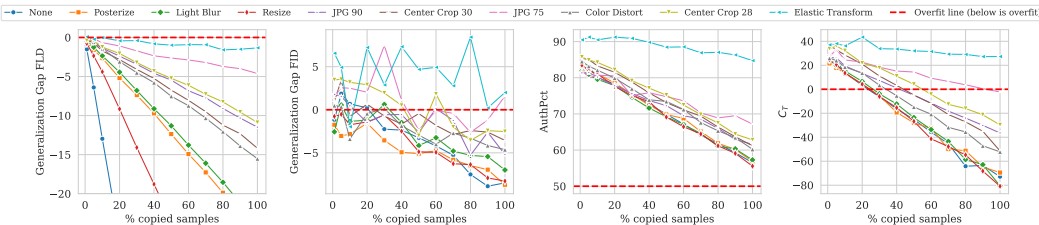

Figure 7: Comparison of various overfitting metrics when $\mathcal{D}_{\text{gen}}$ consists of a combination of transformed generated samples and transformed copies of the train set.

copies of the train set. For each transform in Fig. 7, we start with PFGM++ samples that have had the transform applied and gradually replace them with transformed copies of the training set. As such, sample fidelity/diversity in this "pseudo-generated" set remains roughly constant while overfitting increases.

We then evaluate this set of samples using a variety of metrics designed to detect overfitting. For FLD, we look at the generalization gap $\text{FLD}(\mathcal{D}_{\text{train}}, \mathcal{D}_{\text{gen}}) - \text{FLD}(\mathcal{D}_{\text{test}}, \mathcal{D}_{\text{gen}})$. For FID, we consider the difference $\text{FID}_{\text{train}} - \text{FID}_{\text{test}}$ (where $\text{FID}_{\text{test}}$ is FID using the test set). We use the same amount of samples (10k) for both as FID is biased. $C_T$ [Meehan et al., 2020] is the result of a Mann-Whitney test on the distribution of distances between generated and train samples compared to the distribution of distances between train samples and test samples (negative implies overfit, positive implies underfit). The AuthPct $\in [0, 100]$ is derived from authenticity described in [Alaa et al., 2022] and is simply the percentage of generated samples deemed authentic by their metric.

We find that FLD is the only metric consistently capable of detecting overfitting for all transforms (though the effect is less pronounced for large transforms). The generalization gap of FID oscillates though generally trends in the right direction. However, the magnitude of the gap is small (considering that the FID values using DINOv2 are considerably larger, e.g. the difference between StyleGAN2-ADA and StyleGAN-XL is >60). AuthPct trends in the right direction but seems to only be able to detect a subset of the memorized samples (as it detects none of the sample sets as overfit). Finally, $C_T$ is capable of detecting overfitting for all but the largest transforms (though it does require a significant proportion of copied samples).

### 4.1.4 Interaction effects

We now examine the relative effect of these three axes of evaluation on FLD. For fidelity, we use an increasing blur intensity. For diversity, we take a subset of the generated samples and replace the rest of the sample with copies of this subset. For novelty, we replace generated samples with copies of the training set. Then, in Fig. 8, starting from high quality generated samples, we vary each combination of the 2 axes and observe the effect on FLD through a heatmap.

In summary, Fig. 8 indicates that for FLD, fidelity matters more than diversity which matters more than novelty (with the notable exception of all samples being copied resulting in very high FLD). We argue this ordering is very much aligned with the potential usefulness of a generative model. If samples have poor fidelity, then regardless of their diversity and novelty, they will not be useful. With poor diversity but good fidelity and novelty, removing duplicates yields a useful generative model.

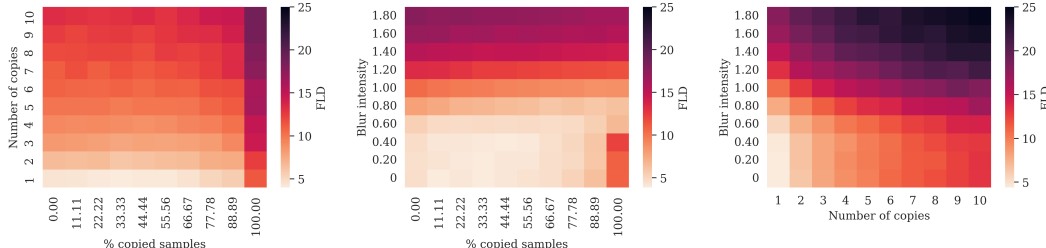

Figure 8: Heatmap of FLD values for all pairs of the three axes of generalization.

Finally, if the model generates copies of the training set, it can still be useful **as long as it generates novel samples as well**.

## 4.2 Comparison of Evaluation of State-of-the-Art Models

Using samples provided by [Stein et al., 2023], we perform a large-scale evaluation of various generative models in Tab. 1 using different metrics on CIFAR10, FFHQ and ImageNet. We find that FLD yields a similar model ranking as FID with some exceptions (for example MHGAN vs BigGAN-Deep on CIFAR10). In addition, we observe that modern generative models are overfitting in a benign way on CIFAR, i.e., there is a noticeable gap between train and test FLD even though the test FLD is reasonably low (indicating good generalization).

| Dataset | Model | FLD | FID | Gen Gap FLD | $C_T$ | AuthPct | Precision | Recall |
|---|---|---|---|---|---|---|---|---|
| CIFAR10 | ACGAN-Mod | 24.22 | 1143.07 | 0.10 | 26.11 | 72.09 | 0.76 | 0.00 |
| | LOGAN | 18.94 | 753.34 | 0.18 | 55.66 | 84.10 | 0.63 | 0.13 |
| | BigGAN-Deep | 9.28 | 203.90 | -0.05 | 55.70 | 88.10 | 0.50 | 0.26 |
| | MHGAN | 8.84 | 231.38 | -0.01 | 47.87 | 86.69 | 0.55 | 0.26 |
| | StyleGAN2-ada | 6.86 | 178.64 | -0.09 | 45.31 | 86.40 | 0.54 | 0.33 |
| | iDDPM-DDIM | 5.63 | 128.57 | -0.33 | 39.65 | 84.60 | 0.57 | 0.55 |
| | StyleGAN-XL | 5.58 | 109.42 | -0.23 | 36.79 | 85.29 | 0.57 | 0.13 |
| | PFGMPP | 4.58 | 80.47 | -0.35 | 32.79 | 83.54 | 0.60 | 0.62 |
| FFHQ256 | Efficient-VDVAE | 9.40 | 465.34 | -0.10 | 32.22 | 86.48 | 0.66 | 0.07 |
| | Projected-GAN | 8.61 | 339.72 | 0.05 | 36.77 | 93.46 | 0.40 | 0.11 |
| | StyleGAN2-ADA | 7.24 | 296.93 | -0.08 | 27.47 | 90.95 | 0.49 | 0.06 |
| | Unleashing | 7.23 | 287.38 | -0.06 | 38.98 | 90.01 | 0.57 | 0.19 |
| | InsGen | 6.20 | 249.91 | -0.08 | 26.06 | 89.74 | 0.52 | 0.12 |
| | StyleGAN-XL | 5.94 | 155.88 | -0.04 | 39.41 | 88.17 | 0.56 | 0.30 |
| | StyleSwin | 5.77 | 200.80 | -0.29 | 32.19 | 87.68 | 0.60 | 0.20 |
| | StyleNAT | 4.69 | 156.38 | -0.12 | 29.38 | 86.35 | 0.62 | 0.30 |
| | LDM | 4.63 | 162.45 | -0.23 | 28.28 | 84.84 | 0.66 | 0.34 |
| ImageNet256 | RQ-Transformer | 11.55 | 212.99 | -0.53 | 125.48 | 86.10 | 0.39 | 0.55 |
| | StyleGAN-XL | 8.46 | 150.27 | -0.40 | 98.69 | 84.10 | 0.43 | 0.26 |
| | GigaGAN | 8.34 | 156.40 | -0.42 | 98.78 | 82.48 | 0.47 | 0.32 |
| | Mask-GIT | 6.74 | 144.23 | -0.63 | 78.97 | 80.02 | 0.48 | 0.44 |
| | LDM | 3.41 | 82.42 | -0.74 | 33.63 | 69.23 | 0.63 | 0.45 |
| | DiT-XL-2 | 1.98 | 62.42 | -0.99 | 22.57 | 65.79 | 0.69 | 0.55 |

Table 1: Summary of performance metrics for various generative models.

## 4.3 Applications of FLD

**Identifying memorized samples**. As discussed in 3.2, we sort the samples generated by various models trained on CIFAR10 according to their respective $\mathcal{O}_j$ and plot samples for different percentiles. As demonstrated in Fig. 9, all models produce a non-negligible amount of very near copies (especially a specific car of which there are multiple copies in the train set). For PFGM++ and StyleGAN-XL near copies exist even at the 1st percentile of $\mathcal{O}_j$ (roughly in line with the findings of [Carlini et al., 2023, Stein et al., 2023]).

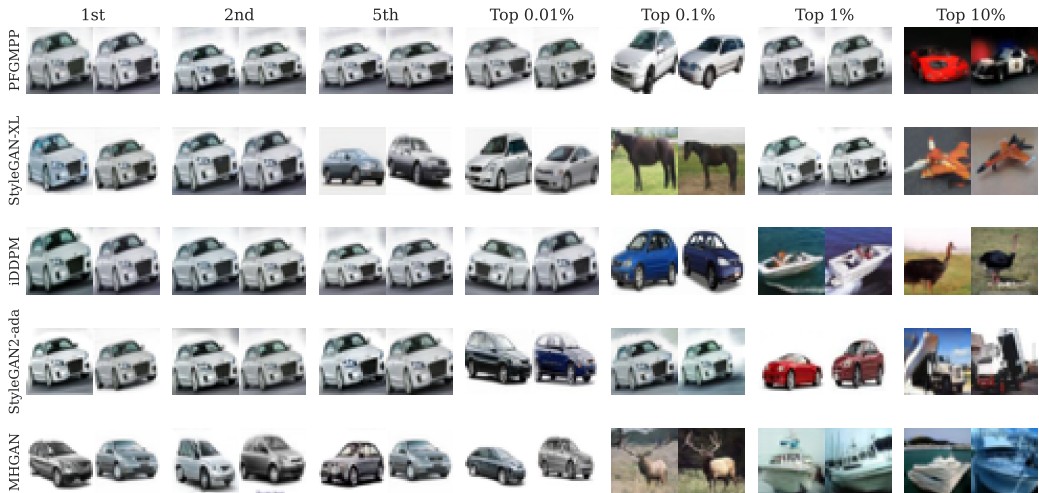

Figure 9: Ranked generated samples for different models from CIFAR10 according to $\mathcal{O}_j$. The left sample is generated, the right one is the nearest sample in the train set (using distances in DINOv2 feature space).

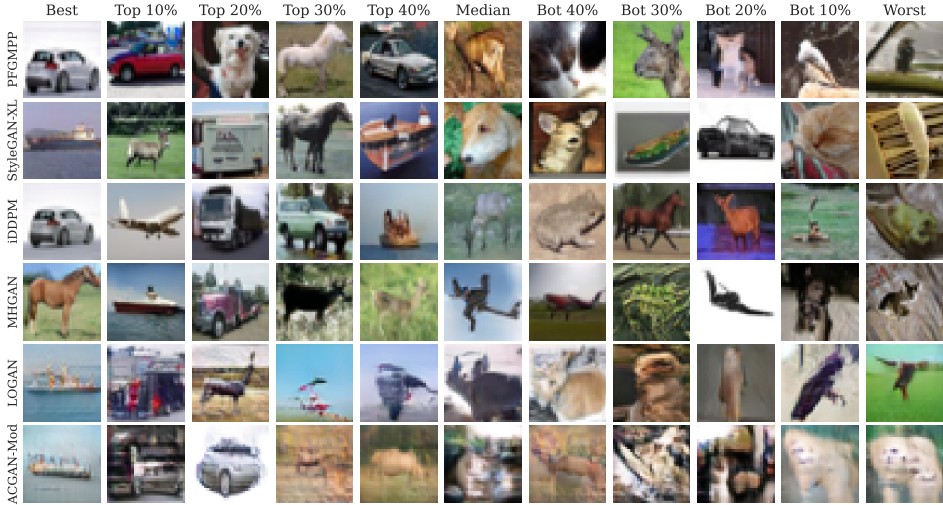

Figure 10: Ranked generated samples for different models from CIFAR10 according to $\mathcal{Q}_j$.

**Evaluating individual sample fidelity**. We repeat the process looking instead at the $\mathcal{Q}_j$ of the models. From Fig. 10, the progression in fidelity of generative models is quite striking with older models such as LOGAN and ACGAN-Mod producing implausible images for all but their best samples. For recent, SOTA generative models, the top half of samples in terms of $\mathcal{Q}_j$ are generally of high quality. However, the bottom half has many examples of poor quality samples that would easily be identified by most humans as being fake. Consequently, even if the ranking is not perfect, filtering generated samples using $\mathcal{Q}_j$ could potentially be beneficial for downstream applications.

## 5   Conclusion

We introduce FLD, a new holistic evaluation metric for deep generative models. FLD is easy to compute, broadly applicable to all generative models, and evaluates generation quality, diversity, and generalization. Moreover, we show that, unlike previous approaches, FLD provides more explainable insights into the overfitting and memorization behavior of trained generative models. We empirically demonstrate both on synthetic and real-world datasets that FLD can diagnose important failure modes such as memorization/overfitting, informing practitioners on the potential limitations of generative models that generate photo-realistic images. While we focused on the domain of natural images, a fertile direction for future work is to extend FLD to other data modalities such as text, audio, or time series and also evaluate conditional generative models.

## Acknowledgements

The authors acknowledge the material support of NVIDIA in the form of computational resources. AJB was generously supported by an IVADO Ph.D. fellowship. Finally, the authors would like to thank Quentin Bertrand, David Dobre, and Alexia Jolicoeur-Martineau for helpful feedback on drafts of this work.

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

# A  Additional Results

## A.1  Number of samples

From Fig. 11, FLD appears to be reliable even when using less reference set samples (similarly to KID Bińkowski et al. [2018]). This is of particular importance for applications in low data regimes or even conditional generation (e.g. the test set of CIFAR10 only contains 1k images for each class). On the other hand, FID highly recommends a minimum of 10k Heusel et al. [2017]. Generally however, the whole training set is used as the value is still biased at 10k and a noticeably lower score can be obtained by using more samples.

As for generated samples, for FID, most papers use 50k as it gives a lower FID than with 10k. We find that using 10k samples for FLD is sufficient for robust evaluation. This is benefit is particularly important given the computational complexity of sampling from modern diffusion models (for example, generating 50k samples often takes hours).

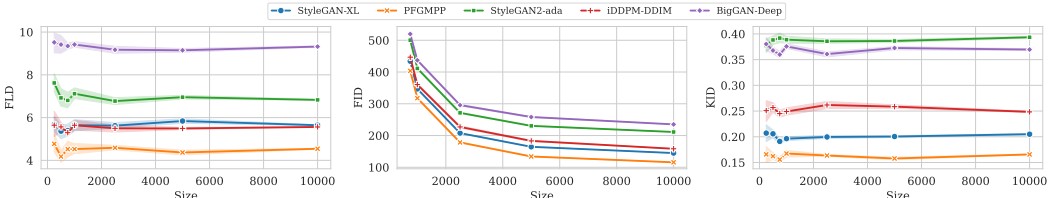

Figure 11: FLD, FID and KID for various sizes of the test set (here, for FID/KID, we use the test set as reference set). Both FLD and KID appear to be relatively consistent, even for smaller test set sizes, while FID is clearly biased.

## A.2  Computational Complexity of FLD

We include the plot of computation time of our score relative to other metrics and find that while it scales linearly with the number of train samples at a faster rate than many metrics, it is still roughly within the same order of magnitude to other metrics and insignificant compared to sample generation time.

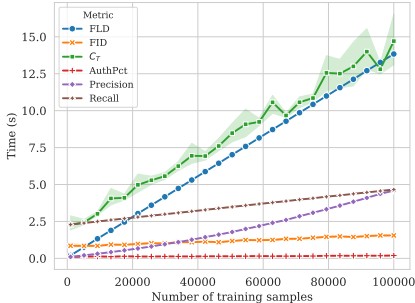

Figure 12: Time taken (on 1xRTX8000) for different metrics as we vary the number of train samples. We solely include the time to compute the metric value once the samples have been generated and mapped to the appropriate feature space (as these two steps are necessary for each metric and are significantly more computationally expensive than the metric computation).

## A.3  Synthetic Experiments

We include a synthetic 2D experiment analyzing how FLD/FID varies as the generated samples go from underfit to overfit. Specifically, we generate 3000 points from the Two Moons dataset using scikit-learn [Pedregosa et al., 2011] with a noise value of 0.1. The first 2000 points are used as

the training set and the last 1000 as test set. We fit a KDE to the train set using bandwidth values varying from $10^{-4}$ to 10 and sample 1000 points as generated samples before computing our score. As expected, FLD decreases initially as the samples begin to match the distribution. However, unlike Test FID, as the samples begin getting too close to the train samples, FLD increases, punishing the model for overfitting.

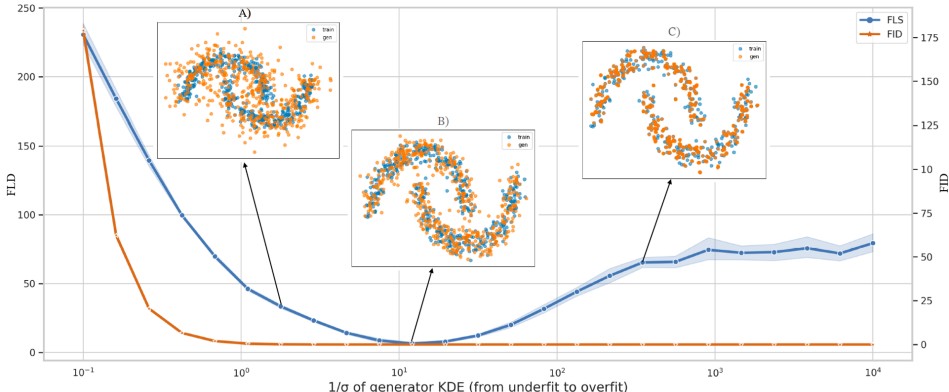

Figure 13: FLD vs. FID for samples from a KDE centered at the train points. As we decrease the bandwidth of the KDE, the generated samples go from A) (underfit e.g. low fidelity resulting in high FID/FLD) to B) (just right, the samples match the distribution but are not overfit to the train samples, yielding low FID/FLD) to C) (overfit, the samples are almost identical to the training samples, yielding low FID but **high FLD**).

# B  Additional Sample Evaluations

Below, we provide additional results of sample evaluations for FFHQ and ImageNet using both $\mathcal{O}_j$ and $\mathcal{Q}_j$.

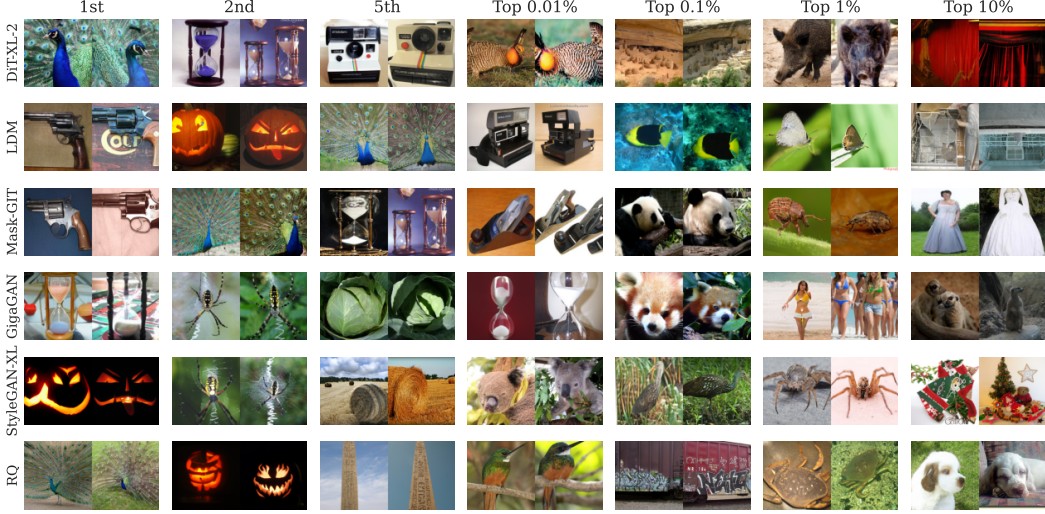

Figure 14: Ranked generated samples for different models on ImageNet according to $\mathcal{O}_j$. The left sample is generated, the right one is the nearest sample in the train set (using distances in DINOv2 feature space).

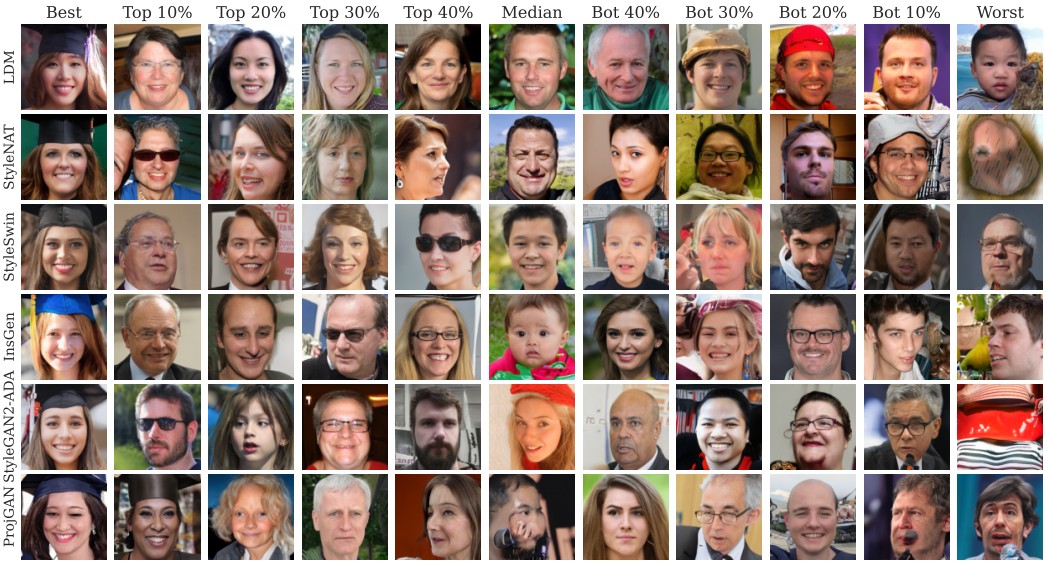

Figure 15: Ranked generated samples for different models on FFHQ according to $\mathcal{Q}_j$.

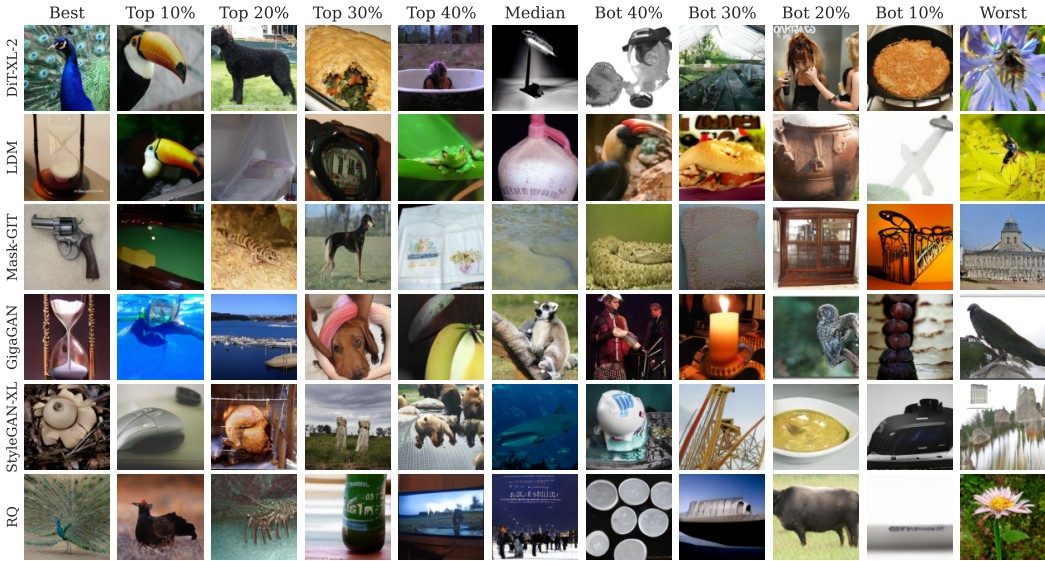

Figure 16: Ranked generated samples for different models on ImageNet according to $\mathcal{Q}_j$.

# C Inception Results

As per the recommendation of Stein et al. [2023], we have updated our experiments to use the DINOv2 feature space instead of Inception-V3. We include below some of the same experimental results with all metrics using the Inception-V3 feature space.

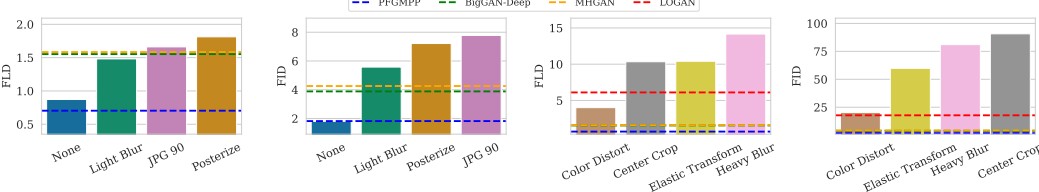

Figure 17: Starting from a set of SOTA samples produced by PFGM++, we replace each sample with a transformed copy. **Left:** Effect of nearly imperceptible transformations on FLD and FID (with corresponding values for various models as reference). **Right:** Effect of large transformations on FLD and FID.

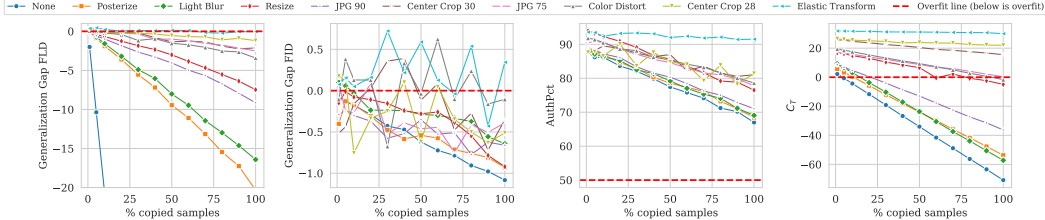

Figure 18: Comparison of various overfitting metrics when $\mathcal{D}_{\text{gen}}$ consists of a combination of transformed generated samples and transformed copies of the train set.

| Dataset | Model | FLD | FID | Gen Gap FLD | $C_T$ | AuthPct | Precision | Recall |
|---|---|---|---|---|---|---|---|---|
| CIFAR10 | ACGAN-Mod | 9.85 | 35.45 | 0.31 | 48.38 | 94.50 | 0.49 | 0.14 |
| | LOGAN | 6.07 | 17.90 | 0.32 | 42.03 | 92.72 | 0.53 | 0.52 |
| | MHGAN | 1.63 | 4.23 | 0.20 | 8.89 | 89.39 | 0.64 | 0.52 |
| | BigGAN-Deep | 1.53 | 3.88 | 0.14 | 5.48 | 87.35 | 0.65 | 0.51 |
| | StyleGAN2-ada | 1.13 | 2.52 | -0.02 | 6.54 | 87.71 | 0.64 | 0.55 |
| | iDDPM-DDIM | 1.03 | 3.30 | -0.03 | 4.00 | 87.31 | 0.68 | 0.58 |
| | StyleGAN-XL | 0.85 | 1.88 | 0.13 | 1.80 | 86.30 | 0.67 | 0.41 |
| | PFGMPP | 0.79 | 1.81 | 0.06 | 4.19 | 87.09 | 0.68 | 0.61 |
| FFHQ256 | Efficient-VDVAE | 3.73 | 34.70 | -0.82 | 2.75 | 85.75 | 0.73 | 0.10 |
| | Unleashing | 1.62 | 8.92 | -0.51 | 7.31 | 87.94 | 0.66 | 0.46 |
| | Projected-GAN | 1.08 | 4.27 | -0.59 | 3.47 | 88.14 | 0.65 | 0.47 |
| | LDM | 1.00 | 8.00 | -0.60 | 5.07 | 87.95 | 0.70 | 0.40 |
| | StyleGAN2-ADA | 0.98 | 5.27 | -0.69 | -1.16 | 86.26 | 0.69 | 0.35 |
| | StyleGAN-XL | 0.92 | 2.19 | -0.56 | 4.42 | 88.55 | 0.66 | 0.51 |
| | StyleNAT | 0.86 | 2.04 | -0.80 | 3.10 | 87.57 | 0.67 | 0.54 |
| | InsGen | 0.75 | 3.37 | -0.53 | 0.70 | 85.83 | 0.68 | 0.45 |
| | StyleSwin | 0.67 | 2.85 | -0.50 | 3.46 | 87.62 | 0.69 | 0.47 |
| ImageNet256 | RQ-Transformer | 2.79 | 9.41 | -1.18 | 8.26 | 81.78 | 0.63 | 0.56 |
| | GigaGAN | 0.04 | 3.83 | -1.48 | -9.10 | 72.48 | 0.75 | 0.45 |
| | StyleGAN-XL | -0.04 | 2.51 | -1.51 | -3.52 | 72.47 | 0.72 | 0.45 |
| | Mask-GIT | -0.34 | 5.36 | -1.95 | -9.13 | 68.31 | 0.78 | 0.46 |
| | DiT-XL-2 | -0.50 | 2.53 | -1.94 | -10.79 | 66.00 | 0.76 | 0.56 |
| | LDM | -0.92 | 3.95 | -1.95 | -21.52 | 63.85 | 0.80 | 0.47 |

Table 2: Summary of performance metrics for various generative models using Inception-V3 features. Interestingly, many models obtain a better NLL than the train set on ImageNet. We hypothesize that this is due to Inception-V3 being trained on ImageNet, resulting in noticeably different activations for the train set.

# D   Method Details

## D.1   Algorithm for MoG

---
**Algorithm 1** Fitting MoGs for FLD
---
**Inputs:** $\mathcal{D}_{\text{train}}, \mathcal{D}_{\text{gen}}, \varphi, \alpha$
**Train**. Fit MoG on $\mathcal{D}_{\text{train}}$ with $\mu = \varphi(\mathcal{D}_{\text{gen}})$
 1: $\sigma^2 = \mathbf{1}$       {// Initialize all bandwidths $\sigma_j^2 = 1$}
 2: $\mathbb{H} = d(\varphi(\mathcal{D}_{\text{train}}), \varphi(\mathcal{D}_{\text{gen}}))$       {// Pre-compute distance matrix.}
 3: **while** $\sigma^2$ not converged **do**
 4:    $\mathcal{L} = -\log p_\sigma(\varphi(\hat{\mathcal{D}}_{\text{train}})|\varphi(\mathcal{D}_{\text{gen}}))$
 5:    $\sigma^2 \leftarrow \sigma^2 - \alpha \nabla_{\sigma^2} \mathcal{L}$       {// Gradient descent on NLL }
 6: **end while**
 7: **return** $\sigma$       {// Return Trained MoG}

---

## D.2   Optimization

Even with the simplifying assumption of a diagonal covariance matrix, to solve Eq. 2 we still need to learn one parameter per Gaussian (of which there are 10000). As the likelihood of a train sample is the result of the $\log$ of the sum of individual Gaussians, the $\sigma_j$ cannot be optimized independently. Thus, we turn to full batch gradient descent using the Adam optimizer with the following hyperparameters:

- 10000 generated samples
- 50 epochs
- $lr = 0.5$
- Initial value for the variance vector: $0$

We modified the original version of Eq. 2 to include a per sample term $\mathcal{L}_i$ intended to provided a minimum amount of likelihood to each train sample. In practice, we set $\mathcal{L}_i$ to be equivalent to the likelihood of the point $\varphi(x_{\text{train}}^i)$ relative to a Gaussian with identity covariance existing at distance $0.9||\varphi(x_{\text{train}}^i)||_2^2$ from $\varphi(x_{\text{train}}^i)$. We find this helps the optimization process recover low variances in the edge cases where most of the generated samples are copied. Otherwise, non-copied train samples have an exponentially small likelihood assigned to them.

In addition, getting the likelihood assigned by each Gaussian to each point requires computing the distance between each pair $(\mathbf{x}_i^{\text{gen}}, \mathbf{x}_i^{\text{train}})$ which is $O(n^2)$ and time-consuming. Also, as the distances and dimensions are large (at least for our high-dimensional experiments), the exponentiation and summations were often numerically unstable. To address this, we:

- Compute the $O(n^2)$ distance matrix once and store it so as not to recompute it for each step of the optimization procedure.
- Optimize the log variances instead of the variances themselves
- Convert $\sigma_j^{-d}$ to $\exp(-d\log(\sigma_j))$ to be able to take advantage of a numerically stable logsumexp.

From plotting the loss, the variances almost always converged in short order (usually less than 10 steps). While we have no guarantee that these were global minima, when there were exact copies or close to exact copies, 1 would recover very low log variances, as shown in 1.

# E  Experimental Setup

## E.1  Transforms

To assess both quality and overfitting, we use standard torchvision [maintainers and contributors, 2016] image transforms described below:

- **Posterize**: Posterize the image to 5 bits (we found 5 was the lowest we could go while the difference still being mostly imperceptible).
- **Resize**: Resize the image to the size accepted by InceptionV3 using bicubic interpolation which has been shown to be problematic [Parmar et al., 2022].
- **Color Distort**: Apply the color jitter transform described in [maintainers and contributors, 2016] with default parameters.
- **Elastic transform**: Apply the elastic transform described in [maintainers and contributors, 2016] with default parameters.
- **Blur**: Gaussian blur with a $(5, 5)$ kernel. For the light blur, we use a $\sigma$ of $0.5$. For the heavy blur we use $\sigma = 1.4$. For the heatmap, the blur intensity corresponds to the value of $\sigma$
- **Center crop x**: Center crop the image to $(x, x)$ and fill the rest with black.
- **JPG x**: Convert image to a JPG of quality x.

## E.2  Generated samples

For all experiments, we have updated our paper to use the samples provided by Stein et al. [2023] who have generously open-sourced an excellent set of samples from a wide variety of SOTA models on various datasets.

## F Proof of Proposition 1

We now prove our Proposition 1 from the main paper.

**Proposition 1.** *Let $D_{ij} := ||\varphi(\mathbf{x}_j^{gen}) - \varphi(\mathbf{x}_i^{train})||^2$ be the distance between a generated sample and a train sample. Assume $\forall i, j : D_{ij} \leq \hat{D}$ with $\delta_j := \min_i D_{ij}$. Then, for any $l \in \{1, \ldots, m\}$, we have that $\hat{\sigma}_l = O(\delta_l)$ where $\hat{\sigma}^2$ is a solution of Eq. 2.*

*Proof.* We begin by showing that $\hat{\sigma}_l^2 = O(\delta_l^2)$ for $\delta_l$ small enough.

We define the variance vector $\hat{\sigma}$ to be the maximizer of the following equation

$$\hat{\sigma}^2 \in \arg\max_{\sigma^2} \frac{1}{n} \sum_{i=1}^{n} \log\left(\frac{1}{m} \sum_{j=1}^{m} \frac{1}{(\sqrt{2\pi}\sigma_j)^d} \exp\left(\frac{-||\varphi(\mathbf{x}_j^{gen}) - \varphi(\mathbf{x}_i^{train})||^2}{2\sigma_j^2}\right)\right). \tag{7}$$

Without loss of generality, we consider the following shifted and rescaled objective function with the same argmax as Eq. 7

$$f(\sigma) := \sum_{i=1}^{n} \log\left(\frac{1}{m} \sum_{j=1}^{m} \frac{1}{\sigma_j^d} \exp\left(\frac{-D_{ij}^2}{2\sigma_j^2}\right)\right). \tag{8}$$

Then, for any $i = 1, \ldots, n$, we have

$$\max_j \sigma_j^{-d} \exp\left(\frac{-D_{ij}^2}{2\sigma_j^2}\right) \leq \sum_{j=1}^{m} \sigma_j^{-d} \exp\left(\frac{-D_{ij}^2}{2\sigma_j^2}\right) \leq m \max_j \sigma_j^{-d} \exp\left(\frac{-D_{ij}^2}{2\sigma_j^2}\right) \tag{9}$$

and using the fact that $\log$ is an increasing function, we have that for any $\sigma$

$$g(\sigma) := \sum_{i=1}^{n} \max_j \left(-d \log \sigma_j - \frac{D_{ij}^2}{2\sigma_j^2}\right) \leq f(\sigma) \leq g(\sigma) + n \log(m) \tag{10}$$

Now, let $\sigma^* \in \arg\max g(\sigma)$ and take $l \in \{1, \ldots, m\}$. We now show that there exists a $\bar{\delta}$ such that for $\delta_l < \bar{\delta}$, $\sigma_l^* = \frac{\delta_l}{\sqrt{d}}$ and appears exactly once in the sum $g(\sigma^*)$.

Suppose $\sigma_l^*$ appears in $k$ terms (WLOG we can assume they are the first $k$ terms). We can then break down $g(\sigma^*)$ as follows,

$$g_k(\sigma^*) = \sum_{i=1}^{k} \left(-d \log \sigma_l^* - \frac{D_{il}^2}{2\sigma_l^{*2}}\right) + \sum_{i=k+1}^{n} \max_{j \neq l} \left(-d \log \sigma_j^* - \frac{D_{ij}^2}{2\sigma_j^{*2}}\right) \tag{11}$$

and define

$$C := \sup_\sigma \sum_{i=k+1}^{n} \max_{j \neq l} \left| -d \log \sigma_j - \frac{D_{ij}^2}{2\sigma_j^2} \right| < +\infty. \tag{12}$$

We then examine the following three cases:

- $k = 0$: The sum in $g(\sigma^*)$ only consists of $j \neq l$.

$$g_0(\sigma^*) = \sum_{i=1}^{n} \max_{j \neq l} \left(-d \log \sigma_j^* - \frac{D_{ij}^2}{2\sigma_j^{*2}}\right) \tag{13}$$

  which is bounded by Eq. 12.

- $k = 1$: Eq. 11 is maximized with respect to $\sigma_l^*$ for $\sigma_l^* = \frac{D_{1l}}{\sqrt{d}}$. Plugging in to Eq. 11

$$g_1(\sigma^*) = -d \log(D_{1l}) - \frac{d}{2}(\log(d) + 1) + \sum_{i=2}^{n} \max_j \left(-d \log \sigma_j^* - \frac{D_{ij}^2}{2\sigma_j^{*2}}\right) \tag{14}$$

Specifically, if the only term $\sigma_l$ appears in is the one with $D_{1l} = \delta_l$

$$\hat{g}_1(\sigma^*) = -d\log(\delta_l) - \frac{d}{2}(\log(d) + 1) + \sum_{i=2}^{n} \max_j \left( -d\log\sigma_j^* - \frac{D_{ij}^2}{2\sigma_j^{*2}} \right) \quad (15)$$

where the term $-d\log(\delta_l)$ and thus $\hat{g}_{k=1}(\sigma^*)$ can be made arbitrarily large as $\delta_l \to 0$. In contrast, if $\sigma_l^*$ appears for $D_{1l} > \delta_l$ we get a bounded $g_1$ (independently of $\delta_l$).

- $k > 1$: Eq. 11 is maximized with respect to $\sigma_l^*$ for $\sigma_l^* = \sqrt{\frac{\sum_{i=1}^{k} D_{il}^2}{kd}}$. Plugging in to Eq. 11

$$g_k(\sigma^*) = \sum_{i=1}^{k} \left( -d\log\left( \sqrt{\frac{\sum_{i=1}^{k} D_{il}^2}{kd}} \right) - \frac{D_{il}^2}{2\frac{\sum_{i=1}^{k} D_{il}^2}{kd}} \right) \quad (16)$$

$$= -kd\log\left( \sqrt{\frac{\sum_{i=1}^{k} D_{il}^2}{kd}} \right) - \frac{kd}{2} \quad (17)$$

$$\leq -kd\log\left( \sqrt{\frac{(k-1)}{kd}}\tilde{\delta}_l \right) - \frac{kd}{2} \quad (18)$$

and is thus bounded independently of $\delta_l$.

Thus, we conclude that, for $\delta_l$ small enough, $\sigma_l^* = \frac{\delta_l}{\sqrt{d}}$ since $\hat{g}_1(\sigma^*) > \{g_0(\sigma^*), g_1(\sigma^*), g_k(\sigma^*)\}$.

Now, consider $\tilde{\sigma}$ such that $\tilde{\sigma}_l = \sigma_l^*/\sqrt{1-c}$ and $\tilde{\sigma}_j = \sigma_j^*$ otherwise. We now look at $g(\tilde{\sigma})$. From the above

$$g(\tilde{\sigma}) = g(\sigma^*) + \frac{d}{2}(\log(1-c) + c) \quad (19)$$

as long as $\tilde{\sigma}_l$ is found in a single term of the summation in $g(\tilde{\sigma})$ (i.e. $k = 1$). If not, with the same argument as above, $g(\tilde{\sigma})$ is bounded above by a constant $U$ independent of $\delta_l$ and $c$. Thus $\exists 0 < \tilde{\delta} < \bar{\delta}$ such that $\forall \delta_l < \tilde{\delta}$, we have $U + n\log m \leq g(\sigma^*)$. Thus, in both cases $\forall \delta_l < \tilde{\delta}$ and for all $c$ such that $\frac{d}{2}(\log(1-c) + c) < -n\log m$, we have

$$f(\tilde{\sigma}) \leq g(\tilde{\sigma}) + n\log m \quad (20)$$

$$\leq \max[U, g(\sigma^*) + \frac{d}{2}(\log(1-c) + c)] + n\log m \quad (21)$$

$$< \max[U, g(\sigma^*) - n\log m] + n\log m \quad (22)$$

$$\leq g(\sigma^*) \quad (23)$$

$$\leq f(\hat{\sigma}). \quad (24)$$

Since $\frac{d}{2}(\log(1-c) + c)$ is decreasing in $c$ (for $c > 0$) and $\tilde{\sigma}_l$ is increasing in $c$, we must have $\hat{\sigma} < \tilde{\sigma}_j = \frac{\delta_j}{d\sqrt{1-c}}$ where $1 - c = \exp(-\frac{2n+1}{d}\log(m))$. Thus, $\hat{\sigma} = O(\delta_j)$ for the case $\delta_l < \tilde{\delta}$.

For the case $\delta_l \geq \bar{\delta}$, we first show that $\hat{\sigma}_l$ is bounded ($\hat{\sigma}_l \leq M$). Denote the individual likelihood terms of $f(\sigma)$ as follows

$$L_{ij}(\sigma_j) := \frac{1}{\sigma_j^d} \exp\left( \frac{-D_{ij}^2}{2\sigma_j^2} \right). \quad (25)$$

Then $\forall i$, we have that

$$L_{il}(\sigma_l) \leq \frac{1}{\sigma_l^d} \underset{\sigma_l \to \infty}{\to} 0 \quad (26)$$

Thus, since $f(\sigma)$ is increasing in the $L_{ij}(\sigma)$, it suffices to show there is $\sigma_l \leq M$ such that all $L_{il}(\sigma_l)$ are positive. Taking $\sigma_l = 1$, we have that $\forall i : L_{il}(1) \geq e^{-\bar{D}^2/2} > 0$. Thus we can take $M = e^{\frac{D^2}{2d}}$. Hence, for the case $\delta_l \geq \tilde{\delta}$

$$\hat{\sigma}_l \leq \frac{\delta_l}{\bar{\delta}} \hat{\sigma}_l$$
$$\leq \frac{M}{\bar{\delta}} \delta_l.$$

Finally $\hat{\sigma}_l$ is $O(\delta_l)$ by taking $C = \max\{\frac{1}{d\sqrt{1-c}}, \frac{M}{\bar{\delta}}\}$.

$\square$

## G  Feature spaces

Feature spaces play a very important role in our metric and similar metrics (FID, Precision/Recall, etc.). While those produced by InceptionV3 have proved useful, there is work illustrating some potential issues, motivating our switch to DINOv2. As such, while the resulting feature psace has proved adequate for our experiments, it is likely that FLD could provide an even better assessment of generated samples with better feature embeddings. Additionally, while this work has solely focused on the evaluation of generative models in the image domain, there is nothing about FLD indicating it could not be applied to other domains (using appropriate feature embeddings) and we leave this as a fruitful future area of research.

## H  Broader Impact

Generative modeling has the potential to create copies of people's identities or artistic styles without their consent, which can have negative impacts on individuals and communities. While evaluation metrics can ensure the generalizability of generative models, they should not be employed to justify copying an artist's style without their permission. The concept of copyrighting an artistic style is intricate, and while some may argue its challenges and ambiguities, it is vital to acknowledge the impact on the artist whose style is being reproduced. Respecting the rights of individuals and communities whose data is used in creating generative models is crucial to avoid harm or exploitation. As the field of generative modeling progresses, it is imperative to engage in ongoing discussions regarding the ethical implications of these technologies and establish guidelines for their responsible application. Additionally, if evaluation metrics fail to consider the generation of harmful or unethical content, negative societal consequences can emerge. Therefore, it is crucial to consider a diverse range of evaluation metrics that encompass various aspects of model performance while aligning with ethical and societal considerations.

