# OpenReview forum: "Feature Likelihood Divergence: Evaluating the Generalization of Generative Models Using Samples"
_NeurIPS.cc/2023/Conference — NeurIPS 2023 poster_

### Official Review · Reviewer_V3Uo · 2023-07-02

**Soundness:** 2 fair
**Presentation:** 3 good
**Contribution:** 3 good
**Rating:** 7
**Confidence:** 4

**Summary:**

The paper suggests a sample-based method (termed FLS) for evaluating generative models. Similarly to FID and IS, the method uses a pre-trained network (InceptionV3 or CLIP) for image representation. Unlike FID, the proposed metric is based on a variant of KDE - fitting isotropic Gaussians around each generated sample and choosing variance values that maximize the likelihood of a subset of train samples. The metric is then the test likelihood. The authors claim and demonstrate that FLS is good at measuring the fidelity, novelty and diversity of generated samples and specifically has the advantages of being able to detect overfitting and memorization.

**Strengths:**

Significance: The ability to quantitively assess the quality of image generation models is highly important, especially due to the recent success and impact of "generative AI". There hasn't been much progress in this area in recent years - FID was introduced in 2017 and is still the most commonly used evaluation method and the de-facto standard. Works in this area therefore have high potential impact and should be encouraged. The proposed method seems to have advantages over FID (e.g. the ability to detect memorization and over-fitting).

Clarity: The paper is very clearly written and is easy to follow. The motivation is clearly stated and the method is well explained.
(I do however have some comments about missing information, in the following sections)

The proposed method of using a KDE-like approach rather than some divergence seems to be original and well motivated.



**Weaknesses:**

Missing information: It is stated that FLS uses "Inception-V3 or CLIP" feature space, however it wasn't clearly indicated what feature space was actually used in each experiment (at least, I couldn't find this information). In addition, I would expect to see an analysis of the different behavior of these two representations (pros and cons of each).

Additional information that I could not find was the number of samples used for FLS evaluation and how it compares with the number of samples needed for FID. This specifically raised a concern because KDE methods (fitting a Gaussian around each sample) in general require a large number of samples. I would also expect that the ability to detect overfit depends on the number of used samples.


**Questions:**

About the correlation between FLS and Sample Fidelity (4.1). Can the authors explain what causes this correlation and the different behavior than in FID? Is it the embedding network or the different approaches to compare distributions?

Have the authors tried evaluating more recent generative models (e.g. Stable Diffusion) on larger datasets (e.g. LAION)? Showing that FLS can scale to these models and dataset can strengthen the significance of the paper.

The proposed method seems to be somewhat related to the NDB evaluation method proposed in [1].

[1] E Richardson, Y Weiss, On GANs and GMMs, Advances in neural information processing systems, 2018

**Limitations:**

A discussion about limitations of the proposed method seems to be missing. One example of such possible limitation is the ability of FLS to scale up to modern image-generation models (e.g. trained on the LAION dataset), generating a much larger variability of images (can the KDE-like approach capture this diversity ?)

I could not identify a possible negative social impact.

---

> ### Author Rebuttal · Authors · 2023-08-10
>
> We thank the reviewer for their thoughtful comments and feedback and we value the fact that the reviewer felt “the ability to quantitively assess the quality of image generation models is highly important” and that works in this area have a “high potential impact”. We also are thrilled to hear that the reviewer found our motivation as being “clearly stated”; that our paper was “well-written” and that our method was “well-explained” and has “advantages over FID”.
> ### Missing Information
> We thank the reviewer for raising the concern about which feature space was used in our experiments. For our experiments—for a fair comparison with FID which uses Inception-v3 — we also use Inception-v3 except for the ImageNet experiments which use CLIP (due to Inception-v3 being trained on ImageNet).
>
> Recent work has also demonstrated that Dino-V2 is a compelling alternative feature space for images, and we provide results on using this feature space in our 1-page PDF. We will add these results to our appendix and explicitly indicate which of Inception, CLIP or Dino-V2 is used where.
> ### Number of Samples used for FLS
> We thank the reviewer for pointing out the importance of the number of samples used for evaluation. For most of our experiments, we use 10k train/test/gen samples (except Table 1 where we use the full train set and 20k generated samples).
>
> Importantly, in Fig 10, we find that FLS is effective and unbiased even when using less test samples, of particular importance for applications in low data regimes or even conditional generation (e.g. the test set of CIFAR10 only contains 1k images for each class). On the other hand, FID highly recommends a minimum of 10k  [1]. Generally however, the whole training set is used as the value is still biased at 10k and a noticeably lower score can be obtained by using more samples.
>
> Similarly, most papers use 50k generated samples as it gives a lower FID than with 10k. We find that using 10k samples for FLS is sufficient for robust evaluation which is in contrast with the number of generated samples used by FID.
>
> ### Correlation between FLS and Sample FID
> We thank the reviewer for their great question. The difference is not due to embedding networks as both FLS/FID are using Inception-V3 for this experiment. There are two ways the behavior is different. For imperceptible transforms, both are affected due to the imperfectness of the feature space. We posit that these transforms affect FID more as they affect the entire distribution of generated samples. As FID examines the distance between distributions, this distribution-wide shift likely has a larger effect (for example, to the covariance matrices of the Gaussians). Conversely, in FLS, the pairwise distances are only slightly increased in the case of these smaller transforms. For Figure 5, we believe the non-linear effect better matches our intuition about the ranking of models and their use in downstream tasks (more details in the global response).
>
> ### Scaling to Text-Image Datasets
> We value the reviewer's feedback on using FLS to evaluate recent generative models, such as Stable Diffusion on LAION. This is indeed an interesting direction of investigation! However, we believe such experiments are beyond the scope of this current paper as Stable Diffusion and models of this type are multi-modal generative models—i.e. text and image—and this presents fresh challenges. For example, we would need to approximate $p(x|y)$ for an unseen text prompt $y$ and it is not immediately clear how this can be accomplished. Note that this is a different setting than the image conditional experiments we considered in Tab. 2 where $y$ is from a known and finite set of class labels. Furthermore, the datasets we chose to evaluate largely are those examined by prior works on evaluation of non multi-modal generative models. Due to these complexities, we believe LAION and Stable Diffusion type models require an extended investigation which is deserving of its own paper.
>
> ### Method is similar to NDB evaluation
> We thank the reviewer for pointing out this interesting reference which we will include in our updated manuscript. Their method, instead of estimating densities like FLS, performs a statistical test on the number of samples in various Voronoi cells for the training/generated sets. The Voronoi cells are constructed by performing K-means clustering of the training set (all of this is done in the pixel space). As a whole, the process is quite different from FLS.
>
> Nonetheless, it is interesting to note that they also use mixtures of Gaussians but for a completely different purpose (as the generative model).
>
> ### Discussion on Limitations
> The reviewer raises a valuable point by adding more discussion regarding the potential limitations of FLS. We agree with the reviewer that such a discussion can help practitioners effectively use FLS for evaluating their own generative models. As we outlined in our response to Reviewer cRFC we find that FLS is less sensitive to the scenario where there is both a little bit of memorization and the generative model still generalizes well (according to definitions 3.1 and 3.2). However, in cases where there is only memorization FLS still correctly reflects this in its score.
>
> While we acknowledge the reviewer's comment regarding a limitation of FLS to larger-scale datasets like LAION we believe this is not a pertinent limitation (as we argue in our response above). We believe that the linear computational complexity of FLS in the number of samples—i.e. $O(n)$ is in line with the best one can hope for any evaluation metric that operates purely on samples.  We will add a bigger discussion of these points in the main paper.
>
> We thank the reviewer for their time and effort in reviewing our work, and we hope our efforts to clarify the main points and allow the reviewer to consider improving their score.
>
> [1] https://github.com/bioinf-jku/TTUR

---

> > ### Comment · Reviewer_V3Uo · 2023-08-11
> >
> > I thank the authors for the thorough rebuttal. Specifically, the fact that FLS does not require an especially large number of samples and is an unbiased estimator is reassuring. I still think handling the new large generative models and datasets would have increased the impact of the method, but I agree with the authors that this can be future work.

---

### Official Review · Reviewer_HK8L · 2023-07-06

**Soundness:** 3 good
**Presentation:** 3 good
**Contribution:** 3 good
**Rating:** 7
**Confidence:** 4

**Summary:**

The work proposes a new metric (FLS) for image generation tasks that is motivated by the observation that current metrics consider the quality and the diversity of the samples, but not their novelty, and therefore are not penalized by memorization of the training set.
The proposed method instead encompasses all three aspects in a single evaluation.
To compute the score, they extract features from the generated, training, and test images. They then build a mixture of isotropic Gaussians centered around the embeddings of the generated samples. The variances are optimized to maximize the log-likelihood of samples of the training set. FLS is then the likelihood of the test samples.

The properties of the metrics are demonstrated by evaluating a number of well-established GAN models.

**Strengths:**

- The paper is well-motivated. While memorization was not much of an issue for generative models so far, some recent study indicates that this might be the case now with the introduction of new powerful generative models. A holistic evaluation including novelty would be a useful tool going forward.
- The principles behind FLS are well-grounded in already established methods (FID, Precision/Recall) and the changes are sound and motivated.
- FLS is not biased by the number of samples and can therefore be compute on a smaller number of points than FID.
- Experiments are extensive, covering many common use-cases including truncation trick, and show promising results. In particular, FLS correlates well with other metrics when memorization is not an issue and the overfitting detection experiments in the main paper and the supplementary are convincing enough.

**Weaknesses:**

A very important aspect is the reliability of FLS w.r.t the number of evaluation samples. As with FID, in practice, people will use the metric in diverse settings and it is important to know when the metric is reliable and when it isn't. The paper touches upon these questions in Fig. 10 but could be more complete. In particular:
- Since the models we compare display very little overfitting, Fig 10 only evaluates the robustness of FLS on the Quality and Diversity aspects, not on the Novelty aspect. As novelty evaluation is one central selling point of the metric, information about the reliability of the overfitting detection aspects is necessary.
- For the Quality and Diversity aspects, FLS should also be compared to KID, which tackles this same issue of sample size bias.  When evaluating with a smaller number of samples, standard deviation should also be provided to better assess the test size that would be needed in practice. Some intervals are depicted in Fig10 but they are barely visible and not commented on at all.

FLS might be harder to interpret intuitively than FID. There is a chance that it could lead to misuse or misunderstandings down the road.
- In the results presented Fig5, FLS is less linear than FID w.r.t. the % of perturbed data. I would guess this effect is caused by the fact that the variance is adaptive, leading to complex interactions between the samples. This could make the interpretation of the values even less intuitive, as we are more used to linear scales.
- In the same figure, until larger percentages of perturbed data, random rotation, elastic transforms, center crops and bicubic resizes all seem to yield the same FLS, while FID assigns different values to different perturbations. As far as I'm concerned, FID results seem more aligned with my intuitions than FLS.

Other weaknesses:
- FLS might be unsuited to easier tasks. Most well-performing models are nearly indistinguishable by FLS in Table 2 (conditional cifar10).
- The computation of FLS has more steps than FID, making it more cumbersome to use. It is also more compute-demanding when scaling.

**Questions:**

My main concerns are, for the sake of practical usage, about the assessment of the minimum number of samples needed:
- Can the authors provide a reliability assessment for overfitting detection in few-sample settings?
- Can the authors update Fig.10 with KID and standard deviations?
- Accordingly, can the authors provide recommendations in terms of the minimal number of samples for reliable assessment?

Related to Fig5 and the related discussion in Section 4.1:
- the authors mention that “FLS is noticeably less affected”, but it is not that clear since we don't have a scale on which to compare the two metrics. Can the authors explain more explicitly how they derive this observation?

As an additional question, one aspect that can be useful in some cases is adversarial perturbation detection.
- How do the authors anticipate the metric to behave when faced with adversarially perturbated images? Considering it is supposed to be more robust to noise, it might not be affected by such perturbation. This can be a good or bad thing depending on the task to be evaluated, but useful to know regardless.

As a final comment, because the paper is proposing a holistic metric, I believe it might be important for the paper to include, in a separate section, a discussion summarizing good practices and recommendations about when and how to use it.

**Limitations:**

Limitations are adequately discussed in the paper and broader impacts in the supplementary.

---

> ### Author Rebuttal · Authors · 2023-08-10
>
> We thank the reviewer for their time and detailed review of our work. We are glad that the reviewer finds our paper “well-motivated”, “well-grounded”, and that the holistic evaluation of generative models as we do by introducing FLS is a “useful tool going forward”. We also appreciate that the reviewer thinks that our empirical results are supported by “extensive” experiments and that the overfitting detecting experiments are “convincing enough”. We now address the key clarification points grouped by theme.
>
> ### Effect of dataset size on FLS (and comparison with KID)
>
> >For the Quality and Diversity aspects, FLS should also be compared to KID, which tackles this same issue of sample size bias.
>
> We have run the experiment and will include an updated version of Fig.10 with KID in the final version of the paper and a complete description of the figure (currently, the confidence intervals are 95% confidence intervals of the bootstrapped distribution). They are very small, essentially invisible for FID. The KID results can be summarized as follows:
> - KID exhibits little to no bias at all dataset sizes but more variance than FLS
> - FLS exhibits some bias at very small dataset sizes (<500) and more variance than FID.
> KID correctly addresses the sample size bias issue of FID but still has the same problem of FID when it comes to detecting overfitting.
>
> ### FLS might be harder to interpret intuitively than FID.
>
> > In [...] Fig5, FLS is less linear than FID w.r.t. the % of perturbed data.
>
> We agree that this is an important point and believe it a non-linear relationship makes sense given the x-axis of the figure. Specifically, a model that produces perfect samples 10% of the time and poor samples 90% of the time has learned to generalize significantly better and is more useful than one that produces poor samples 100% of the time. This argument is detailed further in the global response.
> ### FLS might be unsuited to easier tasks.
> > Most well-performing models are nearly indistinguishable by FLS in Table 2 (conditional cifar10).
>
> We thank the reviewer for pointing out this mistake on our end. The Table 2 in the paper contains unnormalized NLLs (i.e. before subtracting C and multiplying 100). When fixed, the table contains differences in value of similar scale to Table 1. We will fix this small error in the final paper.
>
> ### FLS is more expensive computationally
>
> > The computation of FLS has more steps than FID, [...] It is also more compute-demanding when scaling.
>
> In practice, we find that FLS requires slightly more time to compute than FID at large dataset sizes (as shown in a Appendix F) with computation time scaling linearly with the number of samples. Nonetheless, we believe this is a non-issue as the main bottlenecks are mapping samples to the chosen feature space (~10x slower than computing FLS) and producing samples (up to 100x slower for certain diffusion models).
>
> As for ease of use, all code will be available on GitHub with an easy to use interface for practitioners (provide a folder of images, tensor of features or function that produces samples and the FLS is returned).
>
> > Can the authors provide a reliability assessment for overfitting detection in few-sample settings?
>
> We agree with the reviewer that the few-sample setting is an important area for overfitting detection. While we do not currently have a full reliability assessment, in Appendix B.3. we evaluate StyleGAN2-ADA models trained on datasets of varying sizes (500-4000). There, we find that the % of overfit Gaussians identifies overfitting not detected by other methods.
>
> > Fig5 and the related discussion in Section 4.1: the authors mention that “FLS is noticeably less affected”,[...] Can the authors explain more explicitly how they derive this observation?
>
> Fig. 4 of the rebuttal provides a clearer figure illustrating this behavior. For the transforms which have very little visual effect on humans, the effect on FLS is relatively small whereas for FID, it leads to a significant increase where EDM samples are rated similarly to those produced by SNGAN.
>
> > How do the authors anticipate the metric to behave when faced with adversarially perturbated images?
>
> That is an interesting idea! While we have not tested with any adversarially perturbed image, in light of the above (i.e. the effect of imperceptible transforms), they might be less affected by these perturbations than FID. However, FLS is dependent on the feature embeddings: adversarial perturbations with respect to the Inception-v3 will have a larger effect. This could be tackled by using the features of robust models (e.g. [1]).
>
> > include, [...] a discussion summarizing good practices and recommendations about when and how to use it.
> Thank you for the great suggestion. We will add the following list of good practices to the paper.
> - Keep a separate test set for evaluation. Many generative modeling datasets do not have an explicit test set (FFHQ, LSUN-Bedroom, or LAION [2]).
>   - Recommended to use **a minimum of 1k test samples**, preferably 10k.
>   - Recommended to use **a minimum of 10k generated samples** preferably 50k.
> - Look at FLS but also the generalization gap (difference between Train/Test FLS) and the percentage of overfit Gaussians to identify potentially problematic overfitting.
> - Visually inspect the maximally overfit samples as a means to qualitatively evaluate overfitting behavior.
>
> We thank the reviewer for their valuable feedback and great questions. We hope that our rebuttal fully addresses all the important salient points raised by the reviewer and we kindly ask the reviewer to potentially upgrade their score if the reviewer is satisfied with our responses. We are also more than happy to answer any further questions that arise.
>
> [1] Wang, Zekai, et al. "Better diffusion models further improve adversarial training." 2023.
>
> [2] Schuhmann, Christoph, et al. "Laion-5b: An open large-scale dataset for training next generation image-text models." 2022.

---

> > ### Comment · Reviewer_HK8L · 2023-08-10
> >
> > I thank the authors for the rebuttal and appreciate their consideration of my remarks.
> > My concerns have been convincingly addressed, and I believe the new additions that emerged will be valuable.
> >
> > In particular, I agree with the authors that linearity w.r.t. the percentage of perturbed data is not that important, especially compared to the results in Figure 1 of the rebuttal.
> > Among other things, I also appreciated the recommendations that look reasonable, the updates on KID, and the discussions w.r.t. imperceptible perturbations.
> > From the responses to other reviews, the DINO features experiments in particular would be a very nice addition, providing new insights for generative model evaluation in general beyond this specific method.
> >
> > I will take these elements into account in my final assessment along with the updates from other reviewers before the end of the discussion period.
> >
> > Update: I am raising my rating from WA to Accept

---

### Official Review · Reviewer_SDnT · 2023-07-06

**Soundness:** 3 good
**Presentation:** 3 good
**Contribution:** 2 fair
**Rating:** 6
**Confidence:** 4

**Summary:**

The paper proposes an approach for evaluating generate images by fitting a mixture of Gaussians (MoG) to feature embeddings extracted from the generated and real images. For this, images are first mapped to a feature space (e.g. via an Inception or CLIP image encoder) and then map a Gaussian distribution to each image feature with the mean being centered at the specific image feature. The variance of each Gaussian is then optimized via NLL such that the MoG fits the training examples or a subset thereof.
The MoG can then be used to evaluate image memorization (any Gaussian with near-zero variance will be a memorized sample) and overfitting (if the MoG assigns higher likelihood to the train set than to an unseen test set), as well as sample fidelity and diversity.
The experiments show that the new metric is more robust to small transformations in image space than FID and generally correlates well with FID, precision, and recall.

**Strengths:**

The approach aims to improve upon existing and well-used metrics such as FID and precision+recall. As such, it is easy to calculate, needs minimal human intervention, and scales to large numbers of images.
It also aims to improve the evaluation by detecting overfitting/memorization which many current metrics can't detect. Finally, it can provide more detailed insights into models by evaluating the quality of specific classes individually. The approach also seems to be more robust to using smaller sample sizes (e.g., <5K samples), whereas other metrics such as FID need tens ouf thousands of samples.

**Weaknesses:**

While the approach promises some improvements over other metrics the main improvement seems to be being able to detect overfitting and memorization. However, I am not convinced that feature embeddings are the best way to detect memorization. E.g., looking at the memorized results in Fig 8, those results don't look like memorization to me. Were these obtained based on Inception or CLIP embeddings?

More detailed insights into the performance of specific classes is also already possible with other metrics, e.g., conditional FID.

Aside from that the main other advantage seems to be that the approach is more sample efficient but that is a minor difference since calculating FID and other metrics is usually not the computationally intensive part of model training and evaluation.


**Questions:**

I also wonder about the relationship between this approach and precision+recall. Couldn't precision+recall also be used for detecting memorization and overfitting by applying a similar approach as this one here, applied to the distances already calculated by precision+recall? Would that be simpler? Would the results agree with the results of this metric?

**Limitations:**

My main concern is that the most obvious improvement of the new evaluation method, namely easily detecting memorization and overfitting, may not work as expected since it relies on image embeddings from feature extractors that were not trained for this (neither Inception nor CLIP were trained to detect memorization and instead focus more on high-level features). I would not define image in Fig 8 as memorizations.
Aside from that, other improvements/differences seem marginal compared to existing metrics such as FID and precision+recall and it's not clear to me what exactly makes the new metric better than, e.g., FID.

---

I have adjusted my score to WA based on the author's rebuttal.

---

> ### Author Rebuttal · Authors · 2023-08-10
>
> We thank the reviewer for their time and feedback on our manuscript. We are happy to hear that the reviewer views FLS as “easy to calculate” and that it can scale to a “large number of images”. We also are pleased by the reviewer stating that FLS aims to evaluate the overfitting/memorization behavior of generative models, a fact that “many current metrics can’t detect”. Finally, we thank the reviewer for acknowledging that FLS is “more robust” to smaller sample sizes compared to FID.  We now address the main comments and questions raised by the reviewer.
> ### Detecting memorization using FLS
> > I am not convinced that feature embeddings are the best way to detect memorization […]
>
> We acknowledge the reviewer's healthy skepticism regarding the memorized examples shown in Fig. 8. Our current Fig 8 was obtained using Inception features. We agree that these features may not be ideal and many works have indeed revealed their limitations [1, 2, 3]. While raw L2 distances have shown some promise at detecting memorized examples [4] they are also unreasonably sensitive to imperceptible transformation [5].
>
> We investigate the effect of using a more modern feature extractor in DINO-V2 [3,6] combined with a slightly modified ranking. These results can be found in our 1-page rebuttal PDF in Fig 2. We believe the presented samples are undeniable evidence of copies. We provide more details about this additional experiment in the global response. Specifically, many of these copies (e.g. the car that appears many times in the CIFAR10 training set) exhibit a slightly modified scale/lighting or background shading than the training samples which would lead to a higher pixel-wise distance than their distance in a sufficiently strong feature space. Given this compelling evidence, we believe FLS using DINO-V2 as a feature space is capable of effectively detecting copies.
> ### Using precision+recall to detect memorization
> > Couldn't precision+recall also be used for detecting memorization […] Would the results agree with the results of this metric?
>
> That is an excellent question. FLS bears similarities to a continuous/smooth version of recall.  Unfortunately, directly applying recall as in [8] to the test set (i.e. using the generated samples to get nearest-neighbor (NN) distances) yields the same problem as FID: a model that generates exact copies outperforms SOTA models in recall (0.74 vs 0.70 for EDM).
>
> If we understand correctly, the reviewer is instead referring to using the train samples to get NN distances (similarly to FLS) for the manifold of generated samples.
>
> While this form of recall would be simpler to compute, it presents a key issue addressed by FLS. First, fitting on the train set unfairly rewards bad models: poor quality samples are far from the train set resulting in the construction of balls with larger radii and the engulfing of more test samples. As a result, many models obtain very similar recalls for $k=1$:
> - EDM: 0.41
> - StyleGAN2: 0.4
> - BigGAN-CR: 0.4
> - SNGAN: 0.39
>
> A similar phenomenon occurs for $k > 1$, for example at $k=4$, used in [6]. The binary nature of the recall metric (a sample is either inside or outside the ball) is problematic here. Contrastingly, the likelihood assigned by the MoG in FLS smoothly takes into account the distance of that sample to the MoG. As for precision, it is not immediately clear to us how it could be used to detect overfitting but we are happy to discuss and test other suggestions.
> ### Other Questions
> > More detailed insights into the performance of specific classes is also already possible with other metrics, e.g., conditional FID […] not the computationally intensive part of model training and evaluation.
>
> In certain situations (e.g. small test set for class conditional evaluation), the main bottleneck is not compute, but rather dataset set size. FID is biased and thus requires a large number of samples. For instance, for CIFAR10, practitioners usually consider the FID between 50k generated samples and the training set. Borji [7] mentions that ``A major drawback with FID is its high bias. The sample size to calculate FID has to be large enough (usually above 50K)’’. This is particularly problematic for conditional FID (e.g. CIFAR10 has 5k train + 1k test samples per class while ImageNet has ~1k train + ~150 test), conditional FID would yield a highly biased estimate of the full FID.
> ### Main improvement
> We believe that Fig. 2 in the 1-page PDF which uses the amended memorization metric combined with the DINOv2 feature space convincingly demonstrates the ability to detect memorized samples. In addition, even in Inception space, we provide solid evidence that FLS is capable of detecting overfitting. Fig. 7 shows that FLS is highly affected by the addition of copies and these copies are detected by the percent of overfit gaussians. In App B.1, we show it does so better **even when compared to metrics designed specifically to detect overfitting**.
>
> We thank the reviewer for their time and effort in reviewing our work and we hope the reviewer would kindly consider a fresh evaluation of our work given the main clarifying points outlined above.
>
> [1] Zhou, Sharon, et al. "Hype: A benchmark for human eye perceptual evaluation of generative models." 2019.
>
> [2] Kynkäänniemi, Tuomas, et al. "The Role of ImageNet Classes in Fr\'echet Inception Distance." 2022.
>
> [3] Stein, George, et al. "Exposing flaws of generative model evaluation metrics and their unfair treatment of diffusion models." 2023.
>
> [4] Carlini, Nicholas, et al. "Extracting training data from diffusion models." 2023.
>
> [5] Schäfer,F, et al "Implicit competitive regularization in GANs." 2019.
>
> [6] Oquab, Maxime, et al. "Dinov2: Learning robust visual features without supervision." 2023.
>
> [7] Borji, Ali. "Pros and cons of GAN evaluation measures: New developments." 2022
>
> [8] Kynkäänniemi, Tuomas, et al. "Improved precision and recall metric for assessing generative models." 2019.

---

> > ### Comment · Reviewer_SDnT · 2023-08-15
> > **Thanks**
> >
> > I thank the authors for their extensive response. Using Dino features instead of Inception features does indeed seem to work much better, especially for very close neighbors. The response regarding precision+recall addressed my thoughts and I also especially like Fig 1 of the rebuttal and its implications. I have adjusted my score to WA.

---

### Official Review · Reviewer_cRFc · 2023-07-07

**Soundness:** 3 good
**Presentation:** 3 good
**Contribution:** 3 good
**Rating:** 7
**Confidence:** 4

**Summary:**

This paper addresses the problem that there are currently no sample-based evaluation metrics accounting for the trichotomy between sample fidelity, diversity, and novelty. Likelihood based metrics are not particularly interpretable and sample quality based metrics do not take into account novelty -- they are easy to cheat just by copying the training data. To address these issues, the paper proposes the feature likelihood score (FLS) -- which assesses sample novelty, overfitting, and memorization.

**Strengths:**

* The paper deals with an important problem and proposes an innovative solution.
* The paper provides theoretical guarantees that the proposed FLS score can detect overfitting in Proposition 1.
* The shows evaluations on a variety of datasets -- CIFAR10, ImageNet, LSUN and AFHQ and on a variety of models -- StyleGAN-XL, SNGAN, LOGAN, LSGAN etc.
* The paper is well written and easy to understand.

**Weaknesses:**

* Can the proposed FLS score be extended to Text to image models such as Stable Diffusion or Latent Diffusion.

* The datasets considered such as CIFAR10, ImageNet, LSUN and AFHQ are quite limited in size compared to popular datasets such as LAION. The paper should include an analysis of the effect of dataset size on the ability of FLS to detect generated copies of the training set.

* The paper should also include an analysis of the effect of image resolution on the FLS score -- as the image resolution increases will the FLS score be able to detect copies of the training set images?

* In Figure 7 (right) why does the FID increase for some models when the % copied samples increases.

**Questions:**

* The paper should motivate in more detail why the CIFAR10, ImageNet, LSUN and AFHQ datasets were choosen as the evaluation platform.

* Is the FLS score applicable to text to image models? Can the FLS model detect duplicates in case of text to image models?

* Does the FLS show the same behavior on very large datasets such as LAION as on smaller datasets such as CIFAR10 or ImageNet?

**Limitations:**

The paper does not discuss the limitations of the FLS score in detail. The paper should ideally also failure cases to highlight limitations -- where copied training examples do not degrade the FLS score -- if observed.

---

> ### Author Rebuttal · Authors · 2023-08-10
>
> We thank the reviewer for their time, feedback, and positive appraisal of our work. We are heartened that the reviewer feels that FLS tackles an “important problem and proposes an innovative solution”. We also appreciate that the reviewer finds our evaluation extensive in the number of datasets and models while stating our manuscript is “well written” and “easy to understand”.
>
> ### Extending FLS to Text-to-image
>
> This is a great suggestion! Evaluating multi-modal generative models is an exciting direction for extending FLS. However, multi-modal models present several technical challenges as we need to estimate $P(x|y)$ for unseen $y$ (unrestricted text prompts) of which usually a single $(x,y)$ is available. This is in stark contrast with evaluating class conditional image generative models as we do in Table 2 in our manuscript, as the classes $y$ are known $\textit{apriori}$. Thus while interesting, we believe extending FLS to multi-modal models like latent diffusion deserves its own paper as it requires non-trivial considerations whose solutions are not an immediate application of the conditional FLS.
>
> ### Datasets Considered for Evaluation
>
> We appreciate the reviewer’s concern regarding the datasets used for evaluation. We believe our selection of datasets is relatively standard for image-generative models and is consistent with the primary ones used in the literature of non-multi-modal generative models [1, 2, 3]. Furthermore, such datasets already have pre-trained models, which enable us to perform a large-scale evaluation on a variety of popular generative models (e.g., GANS, Diffusion). Using pre-trained models on standard datasets also ensures fairness and consistency with models evaluated using other measures in the literature. We understand that this motivation may not have been initially evident in the manuscript, and we will update it to highlight this aspect. As for dataset size, in Figure 10  and Appendix B.3, we demonstrate the applicability of FLS in low-data regimes. The effect of FLS versus dataset size. For larger datasets, while we have experiments for ImageNet, LAION is typically not used for non-multi-modal generation. Nonetheless, we believe the aforementioned non-trivial extension of FLS could be used to evaluate generative models trained on LAION and datasets of similar size.
>
>
> ### FLS vs. Image Resolution
>
> We value the reviewer's comment regarding the ability of FLS to detect copies as the image resolution increases. We believe this is a non-issue primarily because FLS operates on the representation space of a pre-trained encoder (e.g., Inception, CLIP, Dino-V2) with a fixed dimensionality regardless of the original input image dimension. Consequently, computing FLS is unaffected by the input image resolution, which we demonstrate on a range of datasets with small resolutions 32x32 for CIFAR10 to AFHQ, consisting of high-quality images of resolution 512x512. For each dataset, we effectively detect overfitting (see Appendix B.2). We hope this allays the reviewer's concern as we argue FLS is independent of image resolution provided a sufficiently rich representation space.
>
> ### Fig 7
>
> The increase in FID is due to the copied samples consisting of transformed versions of the trained samples. For sufficiently strong transformations, the decrease in quality has a more substantial effect than overfitting (considered beneficial by FID). Please see our global response for a detailed discussion on how fidelity, diversity, and novelty affect FLS.
>
> ### Discussion of Limitations
>
> We acknowledge the reviewer's comment on adding a more detailed discussion on the limitations of FLS. We agree with the reviewer that such a discussion can add valuable nuance to our score's proper use and interpretation.
>
> We find that FLS is less impacted if there is a combination of memorization of a subset of training examples and generalization (according to definitions 3.1 and 3.2). This can partially be seen in Figure 7, where a model producing great novel samples 10% of the time and producing copies 90% of the time still gets a reasonnable FLS for a heavy case of memorization. Nonetheless, in such a case, the overfitting is still detected by looking at the gap between the train and test FLS.
> As discussed in Appendix G, our score is heavily influenced by the embedding we consider. We require good embeddings to assess fidelity, diversity, and novelty. For image domains, many efficient pre-trained models exist (CLIP, DINOv2), but this could be a challenge for other modalities such as audio or time series.
> As discussed in Appendix F, our score scales reasonably with the dataset size (i.e., has a similar complexity to its competitors, such as FID). However, it may still be challenging to scale it to very large datasets with billions of training examples.
>
> We will update the paper to highlight such scenarios and include a dedicated limitations section.
>
>
> ### Conclusion and references
>
> We thank the reviewer for their valuable feedback and great questions. We hope that our rebuttal fully addresses all the important salient points raised by the reviewer. We kindly ask the reviewer to consider updating their score if the reviewer is satisfied with our responses. We are also more than happy to answer any further questions that arise. Please do let us know.
>
> [1] Karras, Tero, et al. "Elucidating the design space of diffusion-based generative models." 2022.
>
> [2] Sauer, Axel, Katja Schwarz, and Andreas Geiger. "Stylegan-xl: Scaling stylegan to large diverse datasets." 2022.
>
> [3] Parmar, Gaurav, Richard Zhang, and Jun-Yan Zhu. "On aliased resizing and surprising subtleties in gan evaluation." 2022.

---

> > ### Comment · Reviewer_cRFc · 2023-08-12
> > **Update**
> >
> > Most of my concerns have been addressed, Thanks! Will keep my rating at Accept.

---

### Official Review · Reviewer_euCm · 2023-07-13

**Soundness:** 3 good
**Presentation:** 3 good
**Contribution:** 3 good
**Rating:** 6
**Confidence:** 5

**Summary:**

Limitation of existing evaluation metrics
* Likelihood-based metrics rarely correlated with perceptual fidelity.
* Sample-based metrics are insensitive to overfitting. E.g., FID
* Copycat (=a model randomly outputs training set) outperforms SOTA generators in $\text{FID}_\text{test}$

Proposed evaluation metric: feature likelihood score (FLS)
1. Map generated / train / test images to an embedding (Inception-v3 or CLIP).
1. Initialize isotropic Gaussians centered at the generated samples.
1. Update the variance of above Gaussians to maximize the likelihood of training samples.
1. Compute the NLL of test samples on the Gaussians.

It measures
* Novelty = opposite of Memorization
  * a generated sample is $\delta-$ memorized if NLL of the training set is $\delta-$ lower than the test set on the Gaussian of the generated sample.
* Diversity
  * Empirically shown
* Fidelity
  * Empirically shown

FLS is high when
* The generated samples have poor quality -> Gaussian centers are far from the test set -> high test NLL
* The generated samples do not cover the data manifold -> high test NLL
* The generated samples overfit to the training set -> high test NLL


**Strengths:**

* It measures novelty, fidelity, and diversity of generated samples. (not fully explained below)
* FLS correlates to sample fidelity (actually, corruption) as FID does.
* It is a one-value metric that reflects three aspects. It is easy to rank different models.
* The authors provide the benchmark table of SOTA models.


**Weaknesses:**

1. A super-close literature is missing: [rarity score]
1. Some important definitions are missing. E.g., training/test split, L189 data manifold
1. FLS correlates to sample fidelity not as linearly as FID
1. The mechanism how FLS correlates with sample fidelity/diversity is not described (although straightforward).
1. It is a one-value metric that reflects three aspects. It is difficult to compare different aspects with same value.
1. FLSs for different k and C are not provided.
1. Train FLS and test FLS are not defined.


minor
1. Figure 5 middle shows FID but caption describes it differently.
1. Figure 5 right shows FLS versus Precision but caption describes it FLS versus Recall.


**Questions:**

1. Why should the SOTA generators be better than the copycat?
1. Could you clarify the difference between definition 3.1 and 3.2 other than the margin $\delta$?
1. How do diversity, fidelity, and memorization affect the metric? Is there a metric affects more than another?

To me, Q3 is the most important.

**Limitations:**

Use cases on non-natural images are not shown (mentioned as future work).

---

> ### Author Rebuttal · Authors · 2023-08-10
>
> We want to thank Reviewer euCm for their feedback. We are glad that Reviewer euCm highlighted that a strength of FLS is that it “is a one-value metric that reflects three aspects [Fidelity, Diversity, Novelty],” making it “easy to rank different models.” We now address the specific comments points raised by the reviewer:
>
> ### How do diversity, fidelity, and memorization affect the metric?
> We provide a detailed answer to this question in the global response to the reviewers. We also ran additional experiments that are presented in the 1-page pdf attached.
>
> In summary, Fig.1 indicates that for FLS, fidelity matters more than diversity, which matters more than novelty, with a notable exception when almost all examples are copied where FLS produces among the worst scores. We argue this order is very much aligned with the usefulness of the generative model in a potential downstream task.
>
> ### SOTA generators vs. copycat?
> The “copycat” generative model corresponds to a model that exactly copies the empirical data distribution. Such a model is useless as it cannot generate new samples from the data distribution and is unusable in most downstream tasks—e.g., data augmentation. Conversely, we know that data augmentation using diffusion models provides an improvement in terms of test accuracy [1,2]. A performance metric for generative modeling is useful if it is also a good proxy for the performance of downstream tasks. Hence, SOTA generators should be ranked better than the copycat model.
>
> ### Clarifications on Def 3.1 and 3.2
> In Def. 3.1 we consider the (normal) distribution induced by a **single example** $x_j^{gen}$ and look at the difference between the likelihood of the train set $D_{train}$ and the test set $D_{test}$.
>
> $$ p_{\hat \sigma}(x| x^{gen}_j) := \mathcal{N}( \varphi(x) | \varphi(x_j^{\text{gen}}),\hat \sigma_j^2 I_d)$$
> It corresponds to whether this **individual** point ranks the train set more likely than the test set. (Thus it is a proxy for $x^{gen}_j$ being a memorized point)
>
> In Def. 3.2 we consider the (MoG) distribution $p_{\hat \sigma}$ induced by our sampled generated points $D_{gen}$ (defined in Eq.1).
>
> $$p_{\hat \sigma}(x| D_{\text{gen}}) := \frac{1}{m} \sum_{j=1}^m \mathcal{N}( \varphi(x) | \varphi(x_j^{\text{gen}}),\hat \sigma_j^2 I_d)$$
>
> It corresponds to whether the generated distribution ranks the training set as more likely than the test set—which is a useful measure of detecting the degree of overfitting.
> ### Related work: rarity score
>
> We thank the reviewer for pointing out this recent and relevant reference which we will include in the updated manuscript. Our work complements [3] with a critical difference: FLS evaluates models along the three axes we mention. FLS punishes models that overly memorize, while the rarity score [3] allows models that produce very novel/rare samples to be adequately recognized.
>
> ### Definitions of training/test split, data manifold, train/test FLS
>
> In the definition section, we mention that $D_{train}$ were the samples used to train the generative model, and $D_{test}$ were not used at the training stage. We will clarify this in the revision of the paper.
>
> Regarding “data manifold” L189, we will replace this ill-defined term with “data distribution”.
>
> Train/test FLS refers to the score associated with the likelihood of the training/test set over our density model $p_{hat \sigma}$ (See Eq. 1 and 2 in the submission). Formally,
>
> $$FLS_{train} =  \text{FLS}(D_{\text{train}},D_{\text{gen}}) := - \tfrac{100}{d}\log p_{\hat \sigma}(D_{\text{train}} | D_{\text{gen}}) - C,$$
>
> $$FLS_{test} =  \text{FLS}(D_{\text{test}},D_{\text{gen}}) := - \tfrac{100}{d}\log p_{\hat \sigma}(D_{\text{test}} | D_{\text{gen}}) - C,$$
> We will clarify that point in the revision of the paper.
>
>
> ### FLS correlates with sample fidelity not as linearly as FID
> We thank the reviewer for this critical question. We address it in the global response.
>
> ### Mechanism on FLS and its correlates with sample fidelity/diversity
>
> We describe in L188-191 of the main paper how FLS correlates with sample fidelity and diversity, “Poor sample quality leads to Gaussian centers that are far from the test set and thus a higher NLL. Similarly, a failure to sufficiently cover the data manifold will lead to some test samples yielding very high NLL.” We will clarify the phrasing in the final version.
>
> ### Using FLS (a one-value) metric to reflect three aspects
>
> The field of deep generative modeling has been driven by the use of single metrics for evaluation as it allows ranking models and measuring the progress of the field. For example, FID (fidelity and diversity) has been used to assess the progress on image generation tasks from early GANs such as WGAN-GP to recent diffusion models such as EDM++. As shown in Fig.2 and Tab.1 of our pdf rebuttal, we see the capabilities of these models to memorize and overfit, a facet that is missing current evaluation metrics. That is why we believe FLS can be seen as a holistic score extending the purpose of FID to a third aspect—i.e., novelty.
>
>
> ### FLSs for different k and C are not provided
>
> The $k$s in Fig. 4. are illustrative to show a visual example of density overfitting and are not used to compute FLS. The $C$, on the other hand, is a dataset-dependent constant used to make our score positive (see footnote 2 in main) and does not affect the relative values of FLS between models.
>
> We appreciate the reviewer's feedback on our paper. We believe we have answered all the great points raised by the reviewer in our rebuttal and kindly request a reconsideration of the paper's score. We are also happy to answer any additional questions.
>
> [1] Wang, Zekai, et al. "Better diffusion models further improve adversarial training." 2023
>
> [2] Azizi, Shekoofeh, et al. "Synthetic data from diffusion models improves imagenet classification." 2023
>
> [3] Han, Jiyeon, et al. "Rarity score: A new metric to evaluate the uncommonness of synthesized images." 2022

---

> > ### Comment · Reviewer_euCm · 2023-08-11
> >
> > I appreciate the rebuttal. My minor concerns are addressed.
> >
> > One last thing I care the most is usefulness of the proposed metric related to W5 (one metric) and Q3 (impact of aspects to FLS). It is helpful that the rebuttal pdf reports how FLS changes along fidelity, diversity, and memorization. However, I am not sure if I will use FLS instead of FID unless I want a generative model for data augmentation for discriminative model. Different users have different needs. Users do not know which aspect is affecting FLS. As mentioned by the authors in the rebuttal, FLS might be the same for two models: one model produces 90% good images and 10% bad images with 20% memorization, and another model produces 80% good images and 20% bad images with 10% memorization. As the quality of generated images have been saturated recently, I think we need more specific measures for different aspects.

---

> > > ### Author Response · Authors · 2023-08-12
> > >
> > > We thank the reviewer for their prompt response and willingness to engage in discussion. We believe they bring up two very pertinent points.
> > >
> > > ## Utility of FLS over FID
> > > > I am not sure if I will use FLS instead of FID unless I want a generative model for data augmentation for discriminative models. Different users have different needs.
> > >
> > > We understand the stickiness of FID as a metric and appreciate its utility.  In fact, in most scenarios, we show that FLS largely agrees with FID and yields a similar model ranking (Figure 9.). However, we point out multiple failure modes of FID in the paper and show how they are addressed by FLS.
> > >
> > > - **Heavy bias for smaller samples**: FID displays heavy bias, even up to 50k samples. FLS on the other hand works better with smaller sample sizes which makes it significantly more amenable to evaluating class-conditional generation (where there are often less than 10k samples per class) and for finding problematic classes/issues with conditional alignment.
> > > - **Insensitivity to overfitting/memorization**: FID as it is currently computed (comparing with train samples) decreases with memorization and even when compared to test samples, a model copying the training set gets a SOTA FID.
> > > - **Sensitivity to imperceptible transformations**: Very slightly altered samples can yield considerably worse FID scores. As shown in Fig.3 of the rebuttal PDF, such transforms cause samples from a SOTA model to be rated as worse by FID than those produced by a 5 year old model. As such, small changes in image processing can have disproportionate impacts on FID.
> > >
> > > As well as other issues [1, 2]. We believe it is problematic to continue using a metric with known issues, especially as these issues are becoming more common. As such, we believe that FLS should be favored in most cases and especially when working with smaller datasets. Moreover, note that in our latest experiment (Table 1 of the rebuttal PDF) we observe that FLS with DINOv2 showcases a trend of the superiority of diffusion models in comparison to SOTA GANs (such as StyleGANXL) which was not clear with FID.
> > >
> > > ## Multi-faceted evaluation
> > > >  Different users have different needs. Users do not know which aspect is affecting FLS[...] I think we need more specific measures for different aspects.
> > >
> > > We completely agree that specific metrics (such as precision, recall, rarity score, etc.) are very valuable and should be used to evaluate specific aspects of the best-performing models, but such a diagnosis is complementary to benchmarking and ranking using a single holistic metric. In fact, the use of a single metrics by practitioners is undeniable:
> > >
> > > - **Generative modeling for image data:** FID is by far the most used metric.
> > > - **Machine translation:** Practitioners have been reporting the F-score [3, 4] which trades off precision and recall, or the BLEU score (a precision oriented metric).
> > > - **Supervised classification:** Practitioners often use standard accuracy to evaluate their models even though some datasets have been saturated. Even for unbalanced dataset practitioners often report F1 score [5].
> > >
> > > This can be explained by:
> > > - Comparing two models with respect to several metrics is more challenging since they may be on the Pareto front ($\mathcal{R}^2$ (fidelity, diversity) or $\mathcal{R}^3$ (fidelity, diversity, novelty) are not totally ordered).
> > > - When considering metrics such as precision and recall a model could over-optimize one which corresponds to a pathological behavior while still being on the Pareto front.
> > >
> > >
> > > #### Flexibility of our Method
> > > We would like to point out the flexibility of the feature likelihood methodology. We believe this form of density estimation allows for measuring various aspects of generative model evaluation not possible with FID:
> > > - **Overfitting evaluation:** The % of overfit Gaussians and the generalization gap allow specifically for evaluating the overfitting behavior quantitatively. If the issue affecting FLS is overfitting, it can be picked up by these methods.
> > >   - Ranking of memorized samples provides a way to qualitatively evaluate memorization by the model
> > > - **Fidelity evaluation**: If we estimate the density of the data distribution by using Gaussians centered at the test set, we can then use this to evaluate the likelihood of the generated samples (and thus find high-quality samples, such as in Appendix A.3).
> > >   - The likelihood of the whole dataset could be a quantitative measure of fidelity.
> > >
> > >
> > > [1] Kynkäänniemi, Tuomas, et al. "The Role of ImageNet Classes in Fr\'echet Inception Distance."(2022).
> > >
> > > [2] Parmar, Gaurav, Richard Zhang, and Jun-Yan Zhu. "On aliased resizing and surprising subtleties in gan evaluation." 2022.
> > >
> > > [3] Van Rijsbergen, Cornelis Joost. "Foundation of evaluation." 1974.
> > >
> > > [4] Derczynski, Leon. "Complementarity, F-score, and NLP Evaluation." 2016.
> > >
> > > [5]Wang, Xudong, et al. "Long-tailed recognition by routing diverse distribution-aware experts." 2020.

---

> > > > ### Comment · Reviewer_euCm · 2023-08-15
> > > >
> > > > I appreciate thorough discussion. My concerns are resolved and I am changing my score from BA to WA.
> > > >
> > > > Adding the discussion to the revised version would make the paper stronger.

---

### Author Rebuttal · Authors · 2023-08-10

We thank all reviewers for their thorough reviews and valuable feedback. We are encouraged that they found FLS well-motivated and that the holistic evaluation of generative models has “potential for high impact” (**HK8L, V3Uo**). We also thank the reviewers for viewing our paper as “well written and easy to understand” (**cRFc, V3Uo**). We are also pleased to hear that reviewers found our empirical investigation to contain “extensive experiments” on “SOTA models” on a variety of datasets (**euCM, cRFc, HK8L**). Finally, we appreciate that the reviewers found FLS easy to calculate (**SDnT**) and useful in ranking different models (**euCM**) while being robust to smaller sample sizes in comparison to FID (**SDnT, HK8L**). We now address the main shared concerns, grouped by theme below.

### Additional results on how diversity, fidelity, and novelty affect FLS

To better illustrate the effect of changes in these three aspects of evaluation, we examine how they influence the FLS of 10k CIFAR10 samples generated by EDM G++. We report these results in Fig 1 of the attached 1-page PDF, which shows a plot of the FLS values as heatmaps for each combination of the different axes—i.e., Fidelity vs. Diversity, Novelty vs. Diversity, and Fidelity vs. Novelty. We change the values of fidelity, diversity, and novelty in the following concrete ways:
- **Fidelity:** Increasing severity of Gaussian blur applied to all samples (as measured by the $\sigma$ of the blur).
- **Diversity:** Decreasing diversity by duplicating several times the same sample (e.g., five duplicate samples correspond to replacing the 10k samples with 2k different samples duplicated five times).
- **Novelty:** Increasing the amount of memorized samples (as measured by the % of generated samples replaced by copies of the training set).

In summary, Fig.1 indicates that for FLS, fidelity matters more than diversity which matters more than novelty, with a notable exception when almost all examples are copied where FLS produces among the worst scores. We argue this ordering is very much aligned with the potential usefulness of the generative model in a downstream task:
- If samples have poor fidelity, then regardless of their diversity and novelty, they will not be useful.
- If samples have poor diversity but good fidelity and novelty, removing duplicates yields a useful generative model.
- If there are some copies in the generated sample, it is serviceable as long as it is not the majority.
- However, if the generative model only generates copies, then the generated data is useless (since we already have the training set, we do not need copies of it)

Finally, if a large enough fraction of the generated data point has high fidelity, diversity, and novelty FLS would provide a relatively good score regardless of the other fraction of the generated data. This explains why FLS does not correlates as linearly as FID in Fig 5. We develop this aspect in the following section.


### FLS correlates with sampling fidelity not as linearly as FID

While we agree with Reviewer euCm and HK8L that linear correlation is helpful because it is intuitive, we argue that it is not clear that the most desirable behavior is a linear relationship between our score and **the fraction of perturbed data** (Fig.5 of our submission). In fact, we argue the non-linear relationship is a benefit of our score over FID. For example, consider a model that generates samples with poor fidelity 90% of the time and perfect data 10% of the time. Such a model should be considered significantly better than one that produces poor fidelity samples 100% of the time. The former has potential uses (especially if one can filter out the bad samples), whereas the latter is nearly useless. This aspect is directly captured by FLS—due to its connection to likelihood—which rates the fully copied model as significantly worse (e.g. Fig 5) whereas the more linear relationship found in FID only deems it slightly worse.

Finally, note that in Fig. 1 of the attached pdf, a linear relationship between FLS and the increasing severity of Gaussian blur applied to all samples can be noticed. Thus our score scales linearly with a global change in terms of fidelity (i.e. 100% of the data is similarly corrupted).


### DINO-V2 improves upon Inception features

To address Reviewer SDnT’s comment on the ability of the Inception feature to measure similarities (and thus detect copies from the training set), we conducted additional experiments using a more modern feature extractor in DINO-V2 [1,2], combined with a new ranking. These results can be found in our 1-page rebuttal PDF in Fig 2 and Table 1 (an updated version of Table 1 in the main paper). Visually inspecting the new memorized samples in Fig 2 we believe we have found undeniable evidence for problematic copies that are more visually striking than the ones initially presented in Fig 8. Given this compelling evidence, we believe FLS using DINO-V2 as a feature space is capable of effectively detecting memorized copies.

### Figure 3. FLS is less sensitive to imperceptible perturbations (for the same feature space)

Another benefit of FLS is that it is less sensitive to small, nearly imperceptible perturbations relative to FID. We make this explicit in Fig 3 of our 1-page PDF with Inception-v3 features for transformations corresponding to slight JPG compression, blurring, and posterizing. While FLS is slightly worsened, the FID of EDM samples reaches that of SNGAN.

We thank the reviewers again for their valuable time and feedback. We hope that we address all their questions with this global response and individual responses. We look forward to further discussion.

[1] Oquab, Maxime, et al. "Dinov2: Learning robust visual features without supervision." 2023.

[2] Stein, George, et al. "Exposing flaws of generative model evaluation metrics and their unfair treatment of diffusion models." 2023.

---

### Decision · Program_Chairs · 2023-09-21

**Decision:**

Accept (poster)

**Comment:**

The authors present a novel metric for the evaluation of generative models. The main motivation behind this metric is the fact that current common metrics like FID reward memorization, which this new metric punishes. The authors also show empirically that their metric correlated with fidelity and diversity making it an important tool for comparison between generative models. The authors also addressed all the reviewer's concerns, specifically better showing memorization examples with DINO.